# Refractory Black carbon (rBC) variability in a 47-year West Antarctic Snow and Firn core

Luciano Marquetto[1,2], Susan Kaspari[1], Jefferson Cardia Simões[2,3]

[1] Department of Geological Sciences, Central Washington University, Ellensburg, Washington ZIP Code 98926– USA

[2] Centro Polar e Climático, Universidade Federal do Rio Grande do Sul, Av, Bento Gonçalves 9500, Porto Alegre, Rio Grande do Sul CEP 91509-900 – Brazil

[3] Climate Change Institute, University of Maine, Orono, Maine 04469-5790 – USA

*Correspondence to*: L. Marquetto (luciano.marquetto@gmail.com)

**Abstract.** Black carbon (BC) is an important climate-forcing agent that affects snow albedo. In this work, we present a record of refractory black carbon (rBC) variability, measured from a 20-meter deep snow and firn core drilled in West Antarctica (79°55'34.6"S, 94°21'13.3"W, 2122 m above sea level) during the 2014-2015 austral summer. This is the highest elevation rBC record from West Antarctica. The core was analyzed using a Single Particle Soot Photometer (SP2) coupled to a CETAC Marin-5 nebulizer. Results show a well-defined seasonality with geometric mean concentrations of 0.015 µg L$^{-1}$ for the wet season (austral summer/fall) and 0.057 µg L$^{-1}$ for the dry season (austral winter/spring). The core was dated to 47 years (1968-2015) using rBC seasonality as the main parameter, along with Sodium (Na), Sulfur (S) and Strontium (Sr) variations. The annual rBC concentration geometric mean was 0.03 µg L$^{-1}$, the lowest of all rBC cores in Antarctica referenced in this work, while the annual rBC flux was 6.25 µg m$^{-2}$ a$^{-1}$, the lowest flux in West Antarctica rBC records. No long-term trend was observed. Snow albedo reductions at the site due to BC were simulated using SNICAR-online and found to be insignificant (-0.48%) compared to clean snow. Fire spots inventory and BC emission estimates from the Southern Hemisphere suggest Australia and Southern Hemisphere South America as the most probable emission sources of BC to the drilling site, whereas Hysplit model particle transport simulations from 1968 to 2015 support Australia and New Zealand as rBC sources, with limited contributions from South America. Spectral analysis (REDFIT method) of the BC record showed cycles related to the Antarctic Oscillation (AAO) and to El Niño Southern Oscillation (ENSO), but cycles in common with the Amundsen Sea Low (ASL) were not detected. Correlation of rBC records in Antarctica with snow accumulation, elevation and distance to the sea suggests rBC transport to East Antarctica is different from transport to West Antarctica.

## 1 Introduction

Black carbon (BC) is a carbonaceous aerosol formed during incomplete combustion of biomass and fossil fuels, characterized by strong absorption of visible light and resistance to chemical transformation (Petzold et al., 2013), and plays an important role in the climatic system by being able to alter the planetary albedo (McConnell et al., 2007; Ni et al., 2014).

BC-containing aerosols are the species most commonly identified as being short-lived climate forcers, along with methane and ozone (AMAP, 2015). BC particles stay in the atmosphere for just one week to ten days (Bond et al., 2013; Ni et al., 2014), but during that time they change the direct radiative forcing at the top of the atmosphere by absorbing and scattering sunlight, with high spatial and temporal variability on regional scales (Bond et al., 2013). In some parts of the globe, the impact of BC on the climate can be even higher than greenhouse gasses (Bice et al., 2009). Globally BC is estimated to be second only to $CO_2$ in its contribution to climate forcing, with +1.1 W m$^{-2}$ for the industrial era (1750-2005) (Bond et al., 2013; Ramanathan and Carmichael, 2008)

Increases in BC concentrations in the cryosphere since the industrial revolution have been observed, with most studies focusing on the Arctic, the Himalayas, and European glaciers as these ice caps are close to large urban centers and consequently are influenced by these. Antarctica is a pristine environment far from the rest of the world, but BC can still be found in its atmosphere, snow and ice, as shown by early studies (Chýlek et al., 1987, 1992; Warren and Clarke, 1990). Although there are local emissions of BC due to scientific and touristic activities (Casey et al., 2017; Stohl and Sodemann, 2010), Antarctic ice also records Southern Hemisphere (SH) emissions and long-range transport of BC from low and mid-latitudes (Bisiaux et al., 2012a, 2012b; Pasteris et al., 2014), with BC concentrations in Antarctica being linked to biomass burning from South America, Africa and Australia (Arienzo et al., 2017; Koch et al., 2007; Stohl and Sodemann, 2010). Even tropical latitude emissions have a measurable influence on the continent (Fiebig et al., 2009).

Although there are several records of SH paleo-biomass burning, there are only a few publications on BC variability in ice cores from Antarctica. Some of those are focused on centennial-millennial timescales (Arienzo et al., 2017; Chýlek et al., 1992), and others on annual to decadal scales (Bisiaux et al., 2012a, 2012b; Pasteris et al., 2014). More ice core records are needed to understand the spatial variability of BC transport and deposition to Antarctica, as well as to improve general circulation models (Bisiaux et al., 2012b). In this work we present a new West Antarctic high-temporal-resolution rBC snow/firn core record. This record is the highest West Antarctic rBC record produced to date, and contributes to the understanding of BC temporal and spatial variability in Antarctica.

## 2 Site Description and Field Campaign

The core (TT07) was drilled in the 2014-2015 austral summer on the Pine Island Glacier (West Antarctica) at 79°55'34.6"S, 94°21'13.3"W (elevation 2122 m above sea level – a.s.l.), near the Mount Johns Nunatak (located 70 km NE of the drilling site) (Fig. 1) and close to the Institute/Pine Island ice divide. The drilling site was chosen due to its relatively high accumulation rate, which ensures seasonally preserved stratigraphic resolution (Schwanck et al., 2016; Thoen et al., 2018), and due to the region's interesting pattern of atmospheric circulation, originating from the confluence of air masses from the Weddell, Amundsen and Bellingshausen seas (Parish and Bromwich, 2007; Thoen et al., 2018).

The West Antarctic Ice Sheet (WAIS) is lower elevation and has lower coastal slopes than the East Antarctic Ice Sheet (EAIS), which facilitates the intrusion of moisture-rich cyclones to the interior of the continent and the transport of aerosols

inland (Neff and Bertler, 2015; Nicolas and Bromwich, 2011). Katabatic winds are not as strong in the drilling site region as they are in most of West Antarctica, due to the higher site elevation compared to the surroundings (Parish and Bromwich, 2007). Seasonal differences in atmospheric transport have been reported for the TT07 drilling site, with particle trajectories during the austral summer being slow moving and more locally influenced, while during the winter, air trajectories are influenced by oceanic air masses due to strong westerlies. The majority of air masses arrive from the Amundsen Sea and, secondarily, from across the Antarctic Peninsula and Weddell Sea (Schwanck et al., 2017). These are also the preferred pathway for dust particles (Neff and Bertler, 2015).

We used a Mark III auger (Kovacs Enterprises, Inc.) coupled with an electrical drive powered by a generator (kept downwind at a minimum of 30 meters away) to retrieve the core. The Mark III auger recovers cylinders of 7.25cm diameter and up to one meter long. All sections of the core were weighed in the field, packed in polyethylene bags and then stored in high-density styrofoam boxes. These boxes were sent by air to Punta Arenas (Chile), then to a deposit in Bangor (ME, US) for storing and finally to the Central Washington University Ice Core Laboratory (Ellensburg, WA), where it was kept at -18°C in a clean cold room until sub-sampling and analysis.

## 3 Methods

### 3.1 rBC analytical method

We used an extended range Single Particle Soot Photometer (SP2, Droplet Measurement Technologies, Boulder, CO, USA) at the Department of Geological Sciences, Central Washington University (CWU - WA, USA) to analyze our samples. The particle size range detected by the SP2 at CWU is 80-2000 nm mass-equivalent diameter for the incandescent signal, assuming a void-free BC density of 1.8 g cm$^{-3}$ (Moteki and Kondo, 2010).

The SP2 measures the number and size of rBC particles using laser-induced incandescence, and was used in a variety of studies for BC in snow and ice (Bisiaux et al., 2012a, 2012b; Casey et al., 2017; Kaspari et al., 2014, 2015, 2011; McConnell et al., 2007; Osmont et al., 2018a, 2018b). In this work we use the recommended terminology by Petzold et al. (2013) and present results from the SP2 as refractory black carbon (rBC).

As the SP2 was initially designed to analyze rBC from the atmosphere (dry aerosol), a necessary step to run liquid samples is their nebulization before being coupled to the sample inlet of the SP2. For this, we used a CETAC Marin 5, described in detail by Mori et al. (2016). The authors found a good nebulizing efficiency of 50.0 ± 4.4% and no size dependency in the diameter range of 200-2000 nm. Katich et al. (2017) managed to get nebulization efficiencies near 100% with their equipment set up. We calculated the CWU Marin-5 nebulization efficiency to be 68.3 ± 5.9% (1σ) based on the external calibration carried out every working day using Aquadag standards (Marquetto et al., 2020). We found a decrease in nebulization efficiency during the laboratory work period (-0.31% per working day or -13.3% over the 43 working days), but we assume the nebulization efficiency to remain stable between the measurement of the standard and the samples measured

for the day, as Katich et al. (2017). We attribute this decrease to the Marin-5 but do not see any apparent cause. Liquid pump
flow rates were kept constant at $0.14 \pm 0.02$ mL min$^{-1}$ during analysis.

For details of the CWU SP2 internal and external calibration, refer to Marquetto et al. (2020).

### 3.2 Laboratory and vial cleaning

Regular, intensive cleaning was carried out inside the cold room for all surfaces/parts/equipment in contact with the core
using ethanol and laboratory-grade paper tissues. Tyvek suits (DuPont, Wilmington, DE, USA) and sterile plastic gloves
were used at all times in the cold room during the core processing.

Vials used to store the samples (50 mL polypropylene vials) were soaked in Milli-Q water for 24 hours and rinsed three
times. This process was repeated two more times, in a total of three days soaked in Milli-Q water and nine rinses. The vials
were left to dry, covered from direct contact, in the laboratory.

### 3.3 Sample preparation

The sample preparation process consists of removing the outer layers of the core, as these are prone to contamination during
drilling, handling and transport of the core (Tao et al., 2001). In the cold room, we partitioned the 21 sections of the core
longitudinally, using a bandsaw with a meat grade, stainless steel bandsaw blade. For every cutting session, a Milli-Q (MQ)
ice stick, previously prepared, was cut at the beginning, to guarantee a clean blade for the snow and firn core. After cutting
the core in the bandsaw, we hand scraped the resulting snow and firn sticks with a ceramic knife in a laminar flow hood (still
in the cold room) and cut them in 2 – 2.5 cm samples with the same knife (resulting in ~40 samples per section). We stored
the samples in the pre-cleaned 50 mL polypropylene vials and kept them frozen until analysis. Samples were melted at room
temperature or in a tepid bath not exceeding 25°C, sonicated for 15 min, and then analyzed (in less than 1 h after melting).
The resulting rBC concentrations using this subsampling method were compared to subsampling using a continuous melter
system for the first 8 meters of the core, and results for both methods were statistically the same (Marquetto et al., 2020).

From all steps of the sample preparation, the band saw cutting in the cold room proved to be the most prone to contaminate
samples. An intensive decontamination process was carried out during a month, before we could start working with the core
itself. In order to reach acceptable background levels for this step (around 0.02 μg L$^{-1}$), we replaced and modified some
components of the band saw. We replaced the rubber tires for urethane ones; the carbon blade for a meat grade, stainless
steel blade; the original plastic blade guides for ceramic ones; manufactured an acrylic blade guard, as the original plastic
guard was chipping. Before using the new blade, we burned it using a blowtorch and map/pro gas (propylene with <0.5%
propane) to remove any residues/oils present, then cleaned it with ethanol. For detachable parts, a detergent was used,
followed by ethanol and MQ water. For parts inside the cold room, ethanol was used. We also prepared ice sticks of MQ
water to cut in the band saw and help clean the blade.

### 3.4 Whole-system setup

The setup for the system in use at CWU is as it follows: The melted sample is dispensed to the Marin-5 nebulizer by a Reglo Digital peristaltic pump (ISMATEC, Wertheim, Germany) at $0.14 \pm 0.02$ mL min$^{-1}$ and monitored by a TruFlo Sample Monitor (Glass Expansion, Port Melbourne, Australia). The Marin-5 nebulizer receives standard laboratory air at 1,000 sccm (1.000 L min$^{-1}$), regulated by an Alicat Flow Controller (Alicat Scientific, Tucson, AZ, USA) connected to a Drierite Gas Purifier, which removes any moisture or particulates from the air. The nebulizer heating and cooling temperatures are set to

110°C and 5°C, respectively, following (Mori et al., 2016). We used Tygon LFL tubing ID 1.02mm (Saint-Gobain Performance Plastics, France) for sample to nebulizer connection. The SP2 flow is maintained at 120 volumetric cm³ min$^{-1}$ (vccm). YAG laser power for this project stayed constant above 5.0 V.

Samples were analyzed for 5 minutes each. Procedural blanks (MQ water) were run at the beginning and end of every working day, and also every 15-20 samples. Background levels were kept at 0-0.5 particles cm$^{-3}$ (translating to less than 0.01

135 µg L$^{-1}$ rBC concentration), and a 5% HNO3 solution was used for cleaning the tubing and nebulizer when needed. For the SP2 to go back to background levels, only MQ water was used. Peristaltic pump tubing replacement was necessary only once during the process. The limit of detection (LOD) of the method was estimated to be $1.61\times10\text{-}3$ µg L$^{-1}$ based on procedural blanks measured to characterize the instrument detection limit (mean + $3\sigma$, n=30).

Data processing was performed with the SP2 Toolkit 4.200 developed by the Laboratory of Atmospheric Chemistry at Paul

Scherer Institute (PSI), and was used on the scientific data analysis software IGOR Pro version 6.3.

### 3.5 Fire spots and BC emission database

To help define the dating of the core and to investigate potential emission source regions, we compared our results with two different datasets: BC emission estimates from the Global Fire Emission Database version 4s (GFED4s - Van Der Werf et al., 2017) for the SH (SH South America, SH Africa, Australia and Equatorial Asia) and the Australian and Brazilian

satellite programs, that count the fire spots (number of active fires) in Oceania and South America, respectively.

The GFED4s (https://www.globalfiredata.org/data.html) is based on the Carnegie-Ames-Stanford Approach biogeochemical model (Giglio et al., 2013), and has several improvements compared with the earlier version, including burned area and emissions from small fires as these could be substantial at a global scale (Randerson et al., 2012). BC emission estimates are given in $10^9$ g and separated by region of the globe with a spatial resolution of 0.25 degree latitude by a 0.25 degree

longitude. For the Southern Hemisphere, four regions are identified: Southern Hemisphere Africa (SHAF), Southern Hemisphere South America (SHSA), Australia and New Zealand (AUS) and Equatorial Asia (EQAS).

The Sentinel Hotspots (https://www.ga.gov.au/scientific-topics/earth-obs/case-studies/mapping-bushfires) and the Programa Queimadas (http://www.inpe.br/queimadas/) are fire monitoring programs run by the government of Australia (Geoscience Australia) and Brazil (Instituto Nacional de Pesquisas Espaciais - INPE), respectively. Both programs use Moderate

Resolution Imaging Spectroradiometer (MODIS), Advanced Very High-Resolution Radiometer (AVHRR) and Visible

Infrared Radiometer Suite (VIIRS) sensors to detect areas of elevated infrared radiation. The Sentinel Hotspots holds data from 2002 to present, while the Programa Queimadas has a record of fire spots since 1998. The parameter "fire spot" used in both Australian and Brazilian fire monitoring programs do not translate directly to the dimension and intensity of the biomass burning events, but it holds a correlation with burned area (Andela et al., 2017) and thus can be used to help date the core and investigate potential emission sources.

## 3.6 Core dating

Antarctic ice core rBC records from other sites show a well-defined seasonality, with peak concentrations in austral winter-spring (dry season) due to increased biomass burning activity in the SH during this time of the year (Bisiaux et al., 2012b; Pasteris et al., 2014; Sand et al., 2017; Winstrup et al., 2017). Sodium (Na) and strontium (Sr) also peak in the austral dry season (during winter) due to intense atmospheric circulation and transport (Legrand and Mayewski, 1997; Schwanck et al., 2017). Increased marine biogenic activity reflects an increase in sulfur (S) in late austral summer (Schwanck et al., 2017; Sigl et al., 2016). Also, the maxima in the non-sea-salt sulfur to sodium (nssS/Na) ratio is a robust seasonal indicator and peaks around the new year (Arienzo et al. 2017). This parameter helps in the identification of the annual layers more than the Na and S records alone. Non-sea-salt sulfur was calculated using Eq. 3 to 6 from Schwanck et al. (2017) and references therein.

The core was dated by multi-parameter manual layer counting primarily driven by rBC seasonal variability, as this is a reliable parameter for dating in Antarctica (Sigl et al., 2016; Winstrup et al., 2017), and a well defined seasonality has already been observed for Pine Island glacier (Pasteris et al., 2014). We used S, Sr, Na and nssS/Na records from a core drilled 1 meter away as additional parameters to the main counting.. The trace element records goes down only to ~ 6.5 m, so below 6.5 m the ice core is dated using the rBC record. The trace elements were analyzed by the CCI Thermo Scientific ELEMENT 2 ICP- SFMS coupled to an ESI model SC-4 autosampler; working conditions and measurement parameters are described in Schwanck et al. (2016, 2017).

We considered the new year to match the end of what we define as the austral dry season, as this is a reliable tie point in the record due to the abrupt drop in rBC concentrations. Previous studies have demonstrated that rBC deposition occurs in winter/spring, mostly September to December. For example: Arienzo et al. (2017) observed rBC concentrations to peak in September in the WAIS Divide ice core; Winstrup et al., (2017) used annual variations in rBC as the most reliable annual tracer for the Roosevelt Island Climate Evolution (RICE) ice core, stating that rBC tends to peak earlier in the year than January 1st. Pasteris et al. (2014) also corroborates rBC to peak in October and drop after for the Pine Island and Thwaites Glaciers, with lowest values from February to June. Bisiaux et al. (2012b) state that sub-annual rBC concentrations are highly seasonal in the WAIS Divide ice core for the period spanning 1850-2000 - low austral wet season and high austral dry season concentrations - and presented annual picks in the drop in rBC concentrations, as in this work. This is also consistent with the BC emission estimates from GFED4s and the fire spot databases from Australia and South America.

### 3.7 Snow accumulation, rBC concentrations and fluxes

To account for imperfections in the core geometry (and consequently imprecise density measurements), we averaged the core's density profile with the density profile from Schwanck et al. (2016) for a 45 m deep core drilled in the same region of West Antarctica, 850 m away from TT07. We then fitted a quadratic trend line in the average curve and used this trend line instead of the field measurements to calculate the annual snow accumulation, water equivalent (weq) and rBC fluxes. rBC fluxes were calculated by multiplying annual rBC means by annual snow accumulation.

We consider that the frequency distributions of the core rBC concentrations are lognormal, and so we present geometric means and geometric standard deviations as these are more appropriate than arithmetic calculations (Bisiaux et al., 2012a; Limpert et al., 2001). The geometric standard deviation is the multiplicative standard deviation ($\sigma*$), so the 68.3% interval of confidence is calculated as $_{\sigma}min_{conc}$ = geometric mean • geometric standard deviation, and $_{\sigma}max_{conc}$ = geometric mean/geometric standard deviation (Limpert et al., 2001). Also, correlation analysis was carried out using Mann-Kendall's test; we choose it as opposed to Spearman's test as confidence intervals are more reliable in the former (Kendall and Gibbons, 1990; Newson, 2002).

We present our data as austral summer/fall (wet season: January to June) concentrations and austral winter/spring (dry season: July to December) concentrations. Wet/dry season concentrations and annual concentration geometric means and standard deviations were calculated in the raw rBC measurements using the dating carried out to separate years and rBC concentration variations to pinpoint the changes from dry season to wet season and vice-versa. Monthly mean concentrations were calculated by applying a linear interpolation in the raw measurements, resampling the dataset to 12 values per year.

### 3.8 rBC impact on snow albedo

To investigate BC impact on snow albedo we used the Snow, Ice, and Aerosol Radiation (SNICAR) online model (Flanner et al., 2007). We ran the model using the parameters presented in Table 1 with varying rBC concentrations: We used the wet and dry season geomeans, to analyze variations for both seasons, and the highest seasonal geomean found in the core, which occurred in the dry season. As our aim in this paper is BC, we simulated albedo changes considering only the particulate and disregarding any dust/volcanic ash influence. Snow grain size used was based on Gay et al. (2002).

### 3.9 Spectral analysis

In order to investigate periodic oscillations (cycles) in the TT07 core and BC atmospheric transport to the drilling site, we conducted a spectral analysis in the rBC record using the REDFIT procedure described in detail in Schulz and Mudelsee (2002) in the 'Past – Paleontological Statistics' software version 3.25. The spectral analysis is motivated by the observation that the most predictable (regular) behavior of a time series is to be periodic (Ghil et al., 2002). The REDFIT method is a more advanced version of the simple Lomb periodogram, and can be used for evenly and unevenly sampled data. The model

is fit to an AR(1) red noise model, the bandwidth is the spectral resolution given as the width between the -6dB points and confidence levels of 90, 95 and 99% are presented (based on chi2) (Hammer, 2019).

We chose this approach instead of estimation techniques for evenly spaced data (such as the Multitaper method) because interpolation in the time domain inevitably cause bias and alters the estimated spectrum of a time series (Schulz and Mudelsee, 2002). This way, we used the rBC raw measurements (not resampled, only dated by year and separated by dry/wet season).

We compared the rBC spectrum with the El Niño–Southern Oscillation (ENSO), the Antarctic Oscillation (AAO) and the
225 Amundsen Sea Low (ASL) spectra to observe possible influence of these in the rBC variability. While ENSO and AAO are well-known climate drivers, recent studies have shown the ASL has a profound effect on the West Antarctic climate (Hosking et al., 2013, 2016; Turner et al., 2013). We also compared the core records with the GFED4s BC emission estimates and the Satellite fire spots database to look for similarities between the datasets which could suggest BC emission sources to the drilling site. Table 2 shows the dataset used for the spectral analysis.

**3.10 Particle trajectory simulations**

In order to simulate rBC particle trajectories from source areas to the TT07 drilling site, we used the Hybrid Single-Particle Lagrangian Integrated Trajectory v4 model (HYSPLIT - Draxler and Rolph, 2003; Stein et al., 2015), from NOAA. Hysplit is a complete system for computing simple or complex transport and deposition simulations (Stein et al., 2015) that has been used in Antarctica by several authors (Dixon et al., 2011; Markle et al., 2012; Marquetto et al., 2015; Schwanck et al., 2016a,
2017; Sinclair et al., 2010).

We used global reanalysis data from the National Centers for Environmental Prediction (NCEP) and the National Center for Atmospheric Research (NCAR) – the NCEP/NCAR data set – and ran 10-day (240 h) back-trajectories, every 5 days, from 1968 to 2015, at an initial height of 1000 m. We consider 10 days to be an appropriate simulation time as this is the estimated maximum lifetime of BC in the troposphere (IPCC et al., 2013). An initial height of 1000 m was used in order to
240 minimize disturbance from the underlying terrain, but still maintaining a link with the surface wind field (Sinclair et al., 2010). To identify main airflow patterns at the TT07 drilling site, the individual trajectories were separated into dry and wet seasons (depending on day and month of each run) and simulations from each season were grouped into five clusters using the HYSPLIT model's cluster analysis algorithm.

**4 Results and discussion**

**4.1 Dating**

The core was dated to 47 years (1968-2015), and details are presented in Fig. 2. We consider this dating to have ±2 years uncertainty. The first uncertain year is located at 6.18 m (between 2003 and 2002, figure 2a), where S and nssS/Na peak but no full cycle is observed in the rBC record. We did not consider this to be a year, as rBC does not present a full cycle. The

second uncertain year is located at 18.14 m (year 1973, figure 2b) where there is no clear rBC peak but snow accumulation would be anomalously high if considered to be only a year instead of two. We consider this to be an annual pick and consequently two years, as there is no evidence of higher-than-normal snow accumulation in the region for this period (Kaspari et al., 2004).

## 4.2 Core density and annual snow accumulation

The core density (measured in the field) ranged from 0.38 to 0.60 g cm$^{-3}$. Using the corrected density curve obtained from our field measurements and from Schwanck et al. (2016), we calculated that the 20.16 m length core represents 10.37 m weq m (Fig. 3).

Average annual snow accumulation is $0.21 \pm 0.04$ weq m per year, and varies little throughout the record, with an exception of a peak in accumulation of 0.31 weq m in 1971. The average accumulation is similar to what Banta et al. (2008) found for the WAIS Divide ice core for the last centuries ($0.20 \pm 0.03$ weq m year$^{-1}$, elevation 1759 m a.s.l.) and to the higher altitude cores (>1700 m a.s.l.) from Kaspari et al. (2004) (0.18 to 0.23 weq m year$^{-1}$); although the latter work also presents lower altitude cores (1200 to 1600 m a.s.l.) closer to the drilling site with accumulation rates between 0.32 and 0.42 weq m year$^{-1}$.

## 4.3 rBC concentrations and fluxes

In agreement with others studies (Bisiaux et al., 2012a; Pasteris et al., 2014; Sand et al., 2017; Winstrup et al., 2017) we found a well-marked seasonal rBC cycle along the core, with the same pattern of low summer/fall and high winter/spring concentrations (Fig. 4). As we collected our samples in January and the drilling was carried out from the snow surface, our core starts approximately in the 2015 New Year. The core's annual rBC geometric mean concentration was 0.030 µg L$^{-1}$ with a minimum of 0.001 µg L$^{-1}$ and a maximum of 0.080 µg L$^{-1}$. Winter/spring (dry season) concentration geometric mean was 0.057 µg L$^{-1}$, while summer/fall (wet season) concentration geometric mean was 0.001 µg L$^{-1}$. Wet season average concentrations remained constant over time, while dry season average concentrations showed more variation with peak values in 1999 but no apparent trend. The main results from TT07 rBC analysis is summarized in Table 3.

We calculated annual rBC fluxes to account for potential biases in annual rBC concentrations due to changes in snow accumulation rates. Concentrations and fluxes follow a similar pattern along the core, as can be observed in Fig. 5. This means that rBC concentration variability likely reflects variations in BC emissions, transport and deposition at the site instead of reflecting changes in snow accumulation.

## 4.4 Comparison with other rBC records in Antarctica

BC has been studied in Antarctic snow since the late 1980s and early 1990s (Chýlek et al., 1987, 1992; Warren and Clarke, 1990). These initial studies used filter-based methods, which could under- or overestimate BC concentrations due to some analytical artefacts (Soto-García et al., 2011; Torres et al., 2014; Wang et al., 2012). Studies using the SP2 started appearing more than two decades later, aiming at recent snow rBC concentrations (Casey et al., 2017; Khan et al., 2019), near-surface

air (Khan et al., 2018), recent-past ice cores (couple centuries - Bisiaux et al. (2012b, 2012a); Pasteris et al., (2014)) and the past millennia (Arienzo et al., 2017). From these, a few rBC records overlap temporally with the TT07 core presented in this work (Table 4). rBC concentrations are low at all sites (< 0.5 µg L$^{-1}$), thus small differences in concentration from one core to another could result in a 2-3 fold difference in rBC concentrations.

Pasteris et al. (2014) present rBC records from three high accumulation West Antarctic ice cores: Pine Island Glacier, Thwaites Glacier, and from the divide between the two sites (220, 750 and 370 km apart from TT07 core, respectively). The cores presented annual rBC concentrations of 0.22 µg L$^{-1}$ (Pine Island), 0.21 µg L$^{-1}$ (Thwaites) and 0.20 µg L$^{-1}$ (Divide). The lower altitude cores (DIV2010 – 1329 m a.s.l. and PIG2010 – 1593 m a.s.l.) presented almost 1.5 times more snow accumulation than the higher altitude core (THW2010 – 2020 m a.s.l.), and almost 2 times more than TT07. The mean annual rBC concentrations from Pasteris et al. (2014) are almost six times higher than the rBC annual values observed in TT07. Higher rBC concentrations in Pasteris et al. (2014) could be a result of higher accumulation rates, considering that BC is primarily deposited through wet deposition (Flanner et al., 2007). This is discussed later on in this section.

The WAIS Divide rBC record from Bisiaux et al. (2012a) is located 350 km away from TT07, has similar accumulation rates to TT07, and rBC annual concentration 2.7 times higher than annual values from TT07 (0.08 µg L$^{-1}$ at WAIS and 0.03 µg L$^{-1}$ at TT07). The authors observed a steep increase in rBC concentrations in the WAIS core from 1970 to 2001 (~0.06 µg L$^{-1}$ to ~0.11 µg L$^{-1}$) and related this to an increase in fossil fuel consumption and deforestation in the SH. This increasing trend was not observed in the TT07 core, that showed fairly stable annual concentrations and fluxes through time. Although the WAIS Divide core is located almost at the same distance from TT07 as DIV2010, and farther than PIG2010, its snow accumulation rates and rBC annual concentrations are more similar to TT07 than the cores from Pasteris et al. (2014).

The South Pole samples (1120 km from TT07) from Casey et al. (2017) were collected in early austral summer, possibly still reflecting the SH dry season. They present even higher rBC concentrations than Pasteris et al. (2014), although the samples were collected close to the Amundsen-Scott scientific station and even the "clean air sector" can present local influence, particularly in comparison to the TT07 remote site.

Khan et al. (2018) found rBC concentrations in the same order of magnitude as Casey et al. (2017), although the Dry Valleys collection site from Khan et al. (2018) was far from local interference of scientific station activities. The cores from Bisiaux et al. (2012b) (East Antarctica) present the highest elevations from the cited bibliography, and show similar rBC fluxes comparing to TT07, although these fluxes are a result of high rBC concentrations with low accumulation rates in East Antarctica, while the TT07 fluxes are the opposite – high accumulation rates (similar to the WAIS Divide core) with low rBC concentrations.

Figure 6 shows a comparison of the above mentioned rBC records with snow accumulation, elevation and distance from open sea. Distance from the sea influences rBC fluxes in West Antarctica (Arienzo et al., 2017), and was calculated considering the median sea ice extent from 1981 to 2010 for September (Matsuoka et al., 2018), when rBC emissions start to rise in South America/Australia/New Zealand and rBC concentrations begin to rise in West Antarctica (Arienzo et al., 2017; Bisiaux et al., 2012b; Pasteris et al., 2014). We measured the distance from the rBC records to the closest open sea source

(Amundsen sea for West Antarctic records, Lazarev to Cosmonauts sea for NUS0X-X and Mawson sea for Law Dome). We acknowledge this is a simplistic approximation and that the preferred air mass pathways from the sea to the points are not as straightforward, but for the scope of this work we consider this approximation sufficient.

No patterns are clear for both East and West Antarctica, whereas when considering the data from East and West Antarctica separately, opposite trends are observed. In East Antarctica, rBC concentrations have a negative correlation with snow accumulation and positive correlation with elevation and distance to the sea, whereas in West Antarctica rBC concentrations present a positive correlation with snow accumulation and a negative correlation with elevation and distance to the sea. We observed that for East Antarctica, rBC vs. snow accumulation and rBC vs. elevation presented statistically significant correlations ($r^2 = 0.78$, $p < 0.01$ for the former and $r^2 = 0.79$, $p < 0.01$ for the latter). On the other hand, distance from the sea is not signtifically correlated with rBC ($r^2 = 0.52$, $p = 0.06$).

For West Antarctica, relationships are the opposite: positive correlation between rBC concentrations and snow accumulation ($r^2 = 0.69$, $p = 0.08$) and negative correlations between rBC concentrations and elevation/distance from the sea ($r^2 = 0.30$, $p < 0.33$ for the former and $r^2 = 0.79$, $p < 0.05$ for the latter). McMurdo and South Pole points are not considered in this calculation as they likely reflect local contamination instead of long-range transport (Casey et al., 2017; Khan et al., 2018). Bisiaux et al. (2012b) have also observed negative (positive) relationships between rBC concentrations and snow accumulation (elevation) for East Antarctica, although their comparison also included the WAIS Divide point to the dataset.

These opposite trends may indicate differences in rBC transport to East and West Antarctica. While for East Antarctica upper tropospheric transport and dry deposition may be the main controllers of rBC concentrations (Bisiaux et al., 2012b), for West Antarctica rBC concentrations may be modulated by intrusion of air masses from the marine boundary layer. Low elevations in West Antarctica facilitates the intrusion of moisture-rich cyclones and the transport of aerosols inland (Neff and Bertler, 2015; Nicolas and Bromwich, 2011), while the positive relationship between West Antarctica rBC concentrations and snow accumulation may indicate rBC to be primarily deposited through wet deposition, being scavenged along the coastal regions were snow accumulation is higher.

## 4.5 BC impact on snow albedo

To investigate BC impact on snow albedo we used SNICAR-online to simulate three scenarios with the same parameters but varying rBC concentrations: We ran the model using the wet and dry season geomeans and the highest seasonal geomean (0.015, 0.057 and 0.105 µg L$^{-1}$, respectively). Results show that snow albedo reduction in the TT07 site due to BC is very low to non-existent (Table 5). This was already expected considering (observed) albedo reported by Casey et al. (2017): Although significant albedo reductions have been reported in more contaminated zones near the South Pole Station, the authors found a minor to negligible reduction to albedo for the "clean sector" snow.

We note that this albedo reduction occurs only in the austral summer, as the site is located almost at 80°S.

## 4.6 Emission sources and influence of transport on the record

Variability in ice core records reflects variability in BC emissions, atmospheric transport and deposition (Bisiaux et al., 2012a). As BC stays in the atmosphere for a short period of time (7 to 10 days; IPCC et al., (2013), increases in BC emissions would rapidly reflect increases in BC concentrations in snow, and thus comparing the seasonality of the two records may help to elucidate source regions. To this end, we compared BC monthly emissions in the SH (from the GFED4s model) with monthly rBC values from the TT07 record (Fig. 7a). The BC seasonality at TT07, with increasing concentrations in July, a peak in October and minimum values in April-May, is the same as reported for the nearby Pine Island Glacier (Pasteris et al., 2014; Figure 1) indicating that comparing the TT07 seasonality to regional emissions is valid. Some models indicate that the carbonaceous load in the Antarctic troposphere mainly originates from South American emissions (Koch et al., 2007); others recognize both South America and Australia as the main sources (Stohl and Sodemann, 2010). Although Southern Africa has the largest BC emissions in the SH, it is not considered to be a significant contributor to the aerosol load in Antarctica (Li et al., 2008; Neff and Bertler, 2015; Stohl and Sodemann, 2010). Both Australia (Bisiaux et al., 2012a) and South America (Arienzo et al., 2017) have been suggested as sources of BC to West Antarctica. Figure 7a shows the rBC monthly average values for TT07 (1968-2014) and monthly-averaged BC emissions from GFED4s (1997 – 2015) for the four SH emission regions (regions defined in GFED4s, see website). rBC in the TT07 starts increasing considerably in July, peaks in October and shows high but decreasing concentrations until December.

African emissions increase and decrease earlier in the year compared with other SH emission sources and with the TT07 BC record (Kendall's tau = 0.30, p = 0.17, n = 12). Equatorial Asia BC emissions increases in August and peaks in September, not reflecting the initial rBC increase in TT07 record (Kendall's tau = 0.33, p = 0.13, n = 12). The increasing trend matches South American emissions, as they start rising in the same period, although peaking in September and dropping significantly after (Kendall's tau = 0.66, p < 0.01, n = 12). At last, Australia and New Zealand emit much less BC than the other three regions (Fig. 7b) but atmospheric circulation favors aerosol transport from there to West Antarctica (Li et al., 2008; Neff and Bertler, 2015). Australian and New Zealand emissions start increasing in August and peak in October, falling later than the other regions (December) (Kendall's tau = 0.85, p < 0.01, n = 12). The correlation coefficients then indicate Australia/New Zealand as the most probable source region for BC at the site for the period studied, followed by SH South America. SH Africa and Equatorial Asia present much weaker correlations, which likely indicates these two regions do not contribute substantially to the rBC flux to the TT07 site. This is consistent with previous research (Arienzo et al., 2017; Bisiaux et al., 2012a).

There is a small increase in Australian emissions earlier in the year (May) that is not observed in the TT07 rBC monthly averages. This difference could be associated with the seasonal difference in particulate transport to Antarctica in winter/summer (Hara et al., 2008; Schwanck et al., 2017; Stohl and Sodemann, 2010).

## 4.7 Spectral analysis

We investigated further the possible emission sources and transport influences to the site using the REDFIT spectral analysis. We compared the rBC record with ENSO, AAO and ASL indexes (Fig. 8). This investigation would give information about the effect of local to regional changes in atmospheric circulation on the BC records (Bisiaux et al., 2012a).

The TT07 rBC spectrum showed significant cycles in the 6-year band (AR1 Confidence Interval > 90%) and in the 2-year band (AR1 CI ~90%). Intra annual cycles in the 0.6 and 0.5 frequencies were also observed at a 95% confidence interval. Comparing the TT07 rBC record spectrum with the GFED4s and fire spots spectra, we identified similar periodicities only in the Sentinel Hotspots (Australia) record (Fig. 8), more specifically in the 2 year band (AR1 CI ~90%), and in the 0.6 year band (AR1 CI >90%). All other spectra (including Programa Queimadas satellite data) showed only well-marked annual

periodicities and intra annual periodicities of 2 and 3 cycles per year (0.5 and 0.3-year bands, not shown). We consider some of these intra annual cycles questionable, as the high-frequency end of the spectrum is often overestimated and can present aliases, 'folded signals' of another frequency process (Mudelsee, 2010; Schulz and Mudelsee, 2002), in this case aliases of the annual cycle at the 0.5 and 0.3 year bands. Due to this, we do not consider 0.5 and 0.3-year cycles to be representative. rBC and AAO present similar cycles (2.1 and 0.6-year bands), as well as rBC and ENSO (2-year band).

Using the multitaper method, Bisiaux et al. (2012a) observed the rBC periodicities for the WAIS Divide ice core and Law Dome (both dated to 1850-2001). Although WAIS is closer to the TT07 drilling site (~350 km), the TT07 core presented similarities with the Law Dome spectrum (in the 6 and 2-year bands, not shown). It is not clear to us what the relation between the two sites could be, as the TT07 site location, annual accumulation and site elevation are more related to the WAIS ice core than to Law Dome (Table 4). Arienzo et al. (2017) used the multitaper method to analyze the WAIS Divide

rBC flux for the period spanning 14k-6k years BP, and found a 6.6 year cycle (AR1 CI = 95%) and a 2.3 year cycle (AR1 CI > 95%), similar to the rBC cycles found in this work; although time scales and methodology used were different. Both Arienzo et al. (2017) and Bisiaux et al. (2012a) attribute the 2.3-year cycle to an indirect effect of the Quasi-Biennial Oscillation (QBO). Although the QBO circulation spans the equator to ~30°, QBO-generated variability can affect Antarctica (Strahan et al., 2015), in which case an upper troposphere/stratospheric component may be important to BC

transport to the continent.

In summary, the spectral analysis suggests Australia and New Zealand as the most probable sources of rBC to the drilling site. Also, rBC seems to be related to the AAO (0.6 and 2-year cycle), to ENSO (2-year cycle) but not to ASL, and similarities between rBC cycles in the TT07 site and the WAIS Divide site have been observed.

## 4.8 Particle trajectory simulations using HYSPLIT

We simulated particle transport during the austral wet and dry seasons as another mean of addressing rBC source areas. We ran the HYSPLIT back-trajectory model every five days from 1968 to 2015, for 10 days each (estimated maximum BC lifetime in the troposphere) and clustered the results in five groups for the wet and dry seasons (Fig. 9).

The majority of simulated air parcels arriving at the drilling site presented a slow-moving trajectory (speed is proportional to trajectory length), reflecting a local/regional influence more than long-range transport from other continents (clusters 3 and 4 in Figure 9). This local/regional influence is observed both in the wet and dry seasons, although during the former the contribution of air masses from the Antarctic Peninsula and across (Weddell Sea) are higher than during the latter. A fast-moving, year-round, continental group is also present (cluster 5), and may partly represent katabatic winds flowing from the continent's higher altitudes (East Antarctica) towards lower-altitude West Antarctica. The strongest contribution of long-range air parcels is from the South Pacific (clusters 1 and 2). These air masses are also fast-moving and present slight seasonal variations, shifting pole wards during the wet season, when they represent 34% of all air parcels, and away from Antarctica during the dry season, when they respond for 22% of all air parcels modelled.

Results from clusters 1 and 2, along with individual trajectories of each cluster (Fig. 10) support our conclusion that Australia and New Zealand are the most probable sources of rBC to the drilling site, considering tropospheric transport. The most visible influence of air parcels from these two countries to the drilling site can be seen in the individual trajectories of cluster 1 (Fig. 10) for both dry and wet season, while for cluster 2 and 4 there are trajectory variations from one season to another. The poleward shift of cluster 1 trajectories in the wet season (Fig. 9) may be a reason why the Australian emissions earlier in the year (May) are not visible in the TT07 rBC record. South American influence to the TT07 drilling site, on the other hand, is restricted to the higher latitude countries (Chile, Argentina), as shown in the individual trajectories of clusters 2, 3 and 5 (Fig. 10). This suggests that South American fires are not significant contributors to the rBC concentrations observed at the TT07 site when considering only tropospheric transport.

## 5 Conclusions

BC in Antarctica has been studied only in the recent decades, but long-range anthropogenic influences have already been observed (Bisiaux et al., 2012a; Stohl and Sodemann, 2010). Models predict a continued increase in BC emissions from source areas (Bond et al., 2013) and a continued increase in BC flux to the Antarctic region, mostly to the Antarctic Peninsula and West Antarctica (Arienzo et al., 2017). Understanding the spatial variability of BC is then essential to predict BC's future impact on the continent.

We analyzed a 20-meter long snow/firn core from West Antarctica spanning 1968-2015 for rBC. Results show a well-defined seasonal variability in the record, with low (high) concentrations during the Southern Hemisphere wet (dry) season but no long-term trend along the 47 years of the core. Snow accumulation remained stable during this period. rBC annual concentrations were found to be the lowest in samples from the recent decades compared to other studies, while rBC annual

fluxes compare with the low values found by Bisiaux et al. (2012b) for high elevation East Antarctica ice cores. Correlations between rBC and snow accumulation, elevation and distance to the sea for East and West Antarctica records indicate rBC transport and deposition might be different for each. SNICAR modeling indicated BC does not affect snow albedo significantly at the site, with a reduction of 0.48% and 0.41% for the highest rBC concentrations found in the core and for dry season geomean concentrations relative to clean snow, respectively. Negligible impact in albedo was observed for wet season geomean concentrations. BC emission estimates, satellite data of fire spots and HYSPLIT particle transport simulations suggest Australia and New Zealand as the main contributors to the rBC present in the TT07. Based on GFED4s emission estimates, SH South America may be a secondary contributor, although this is not supported by spectral analysis results or air mass trajectories. Spectral analysis of the rBC shows  influence of AAO and ENSO periodicities; ASL influences were not detected. This core is the highest elevation rBC core collected in West Antarctica and its low BC concentrations compared to previous studies indicates spatial variability in the transport and deposition of BC in West Antarctica.

**Data availability:** TT07 data is available upon request; auxiliary data can be downloaded from respective sources cited along this work.

**Author Contributions:** Luciano Marquetto: conceptualization, investigation (fieldwork, laboratory), formal analysis, writing – original draft; Susan Kaspari: conceptualization, formal analysis, investigation (laboratory), methodology (laboratory), resources (laboratory), supervision, validation, writing – review & editing; Jefferson Cardia Simões: Conceptualization, funding acquisition, investigation (fieldwork), project administration, resources (fieldwork), supervision, writing – review & editing.

**Competing interests:** The authors declare that they have no conflict of interest.

**Acknowledgements:** This research is part of the Brazilian Antarctic Program (PROANTAR) and was financed with funds from the Brazilian National Council for Scientific and Technological Development (CNPq) Split Fellowship Program (no. 200386/2018-2, from the CNPq projects 465680/2014-3 and 442761/2018-0, CAPES project 'INCT da Criosfera' 88887.136384/2017-00 and PROANTAR project 88887.314450/2019-00. We also thank the Centro Polar e Climático (CPC/UFRGS) and the Department of Geological Sciences (CWU) faculty and staff for the support to this work. We also thank authors from Bisiaux et al. (2012a, 2012b), Casey et al. (2017), Khan et al. (2018) and Pasteris et al. (2014) for rBC data availability.

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

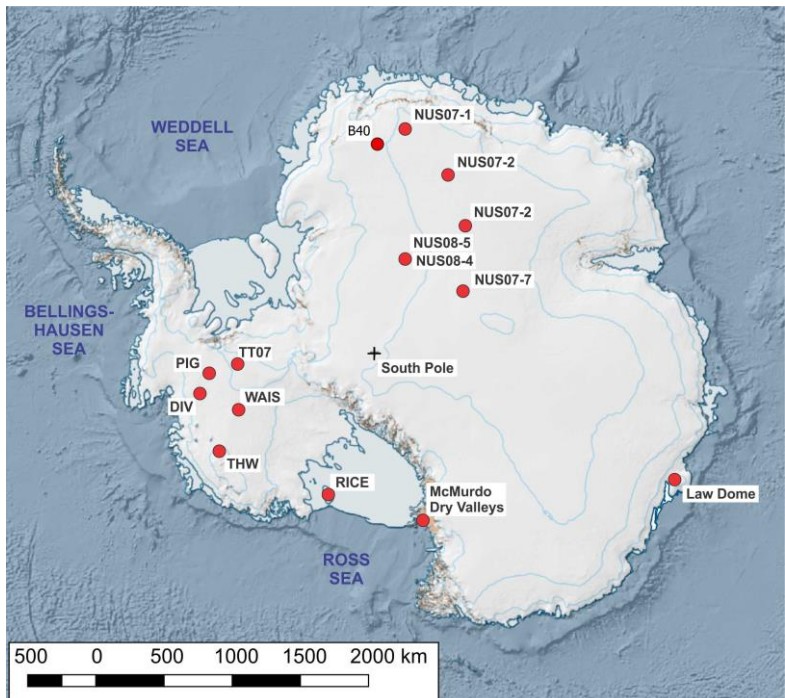

Figure 1. Drilling location for the snow and firn core analyzed in this work (TT07) and other points of interest mentioned in the text. Basemap from the Quantarctica Project (Matsuoka et al., 2018).

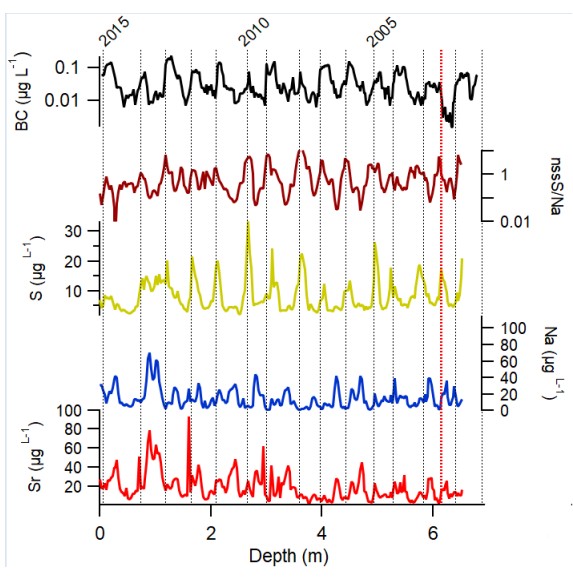

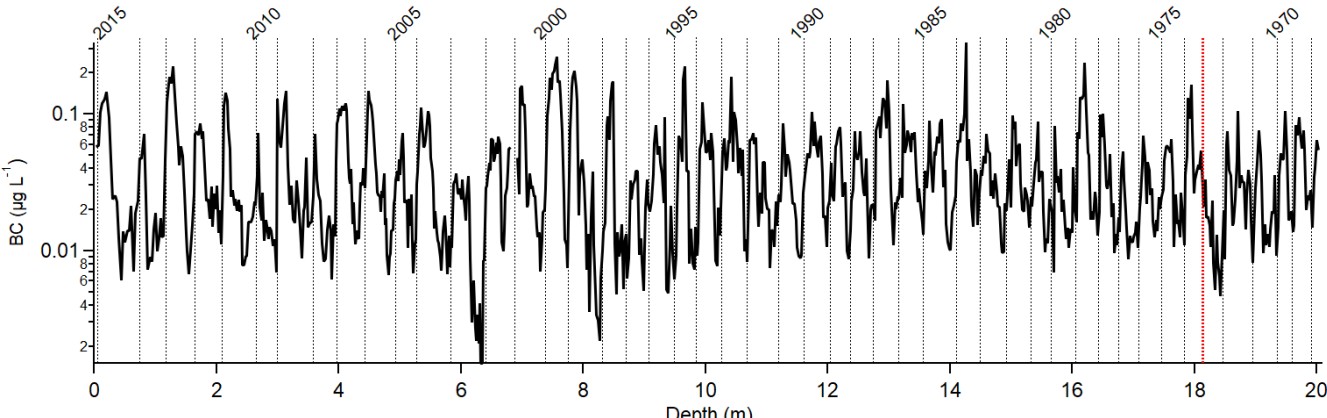

**Figure 2. (a) Dating of the snow and firn core based on rBC and using S, Sr, Na and nssS/Na records from nearby core (see section 3.6) as support for the first 6.5 meters. Dashed lines indicate estimated New Year and red dotted line indicate uncertainty in dating, explained in the text. (b) Dating for the full core (y axis logarithmic). Red dotted line indicates uncertainty in dating, as explaned in the text.**

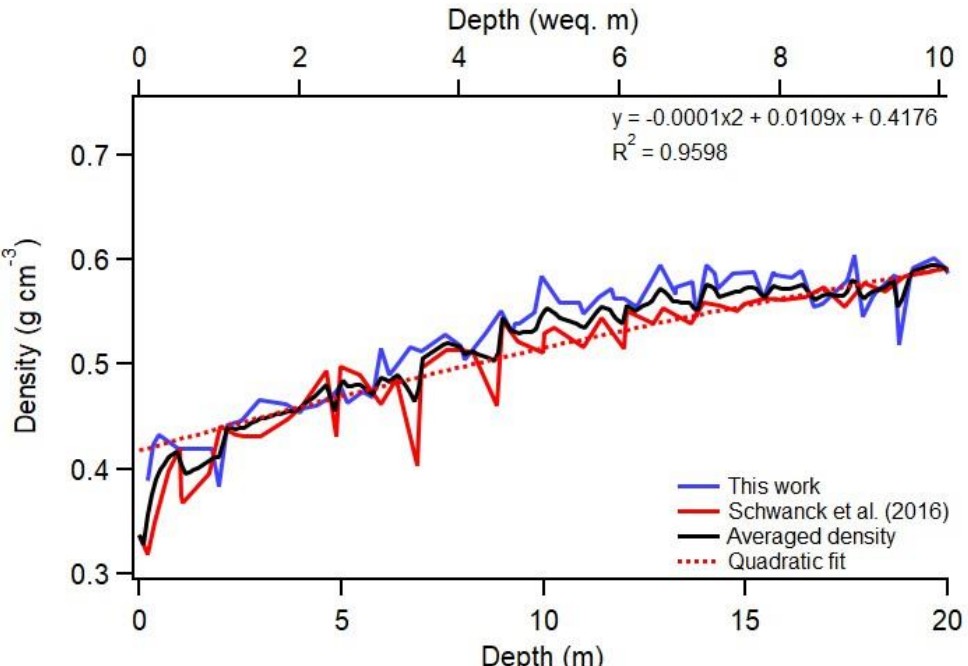

**Figure 3. TT07 density profile (blue). Depth is presented in meters and water equivalent (weq) meters. The quadratic fit was calculated from the average density profile (black) from this work and from Schwanck et al. (2016).**

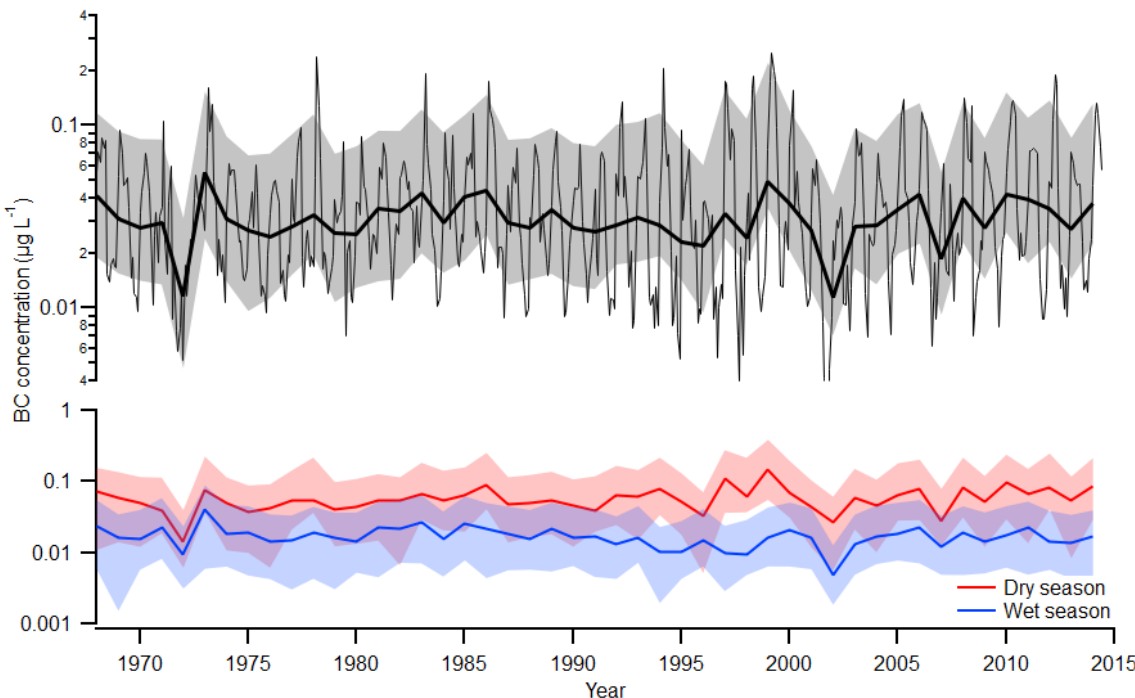

**Figure 4. (top) rBC concentrations for the entire core. The black thick line represents annual averages, while the gray line represents monthly values. Note the y axis scale is logarithmic. (bottom) Dry season and wet season average concentrations per year.**

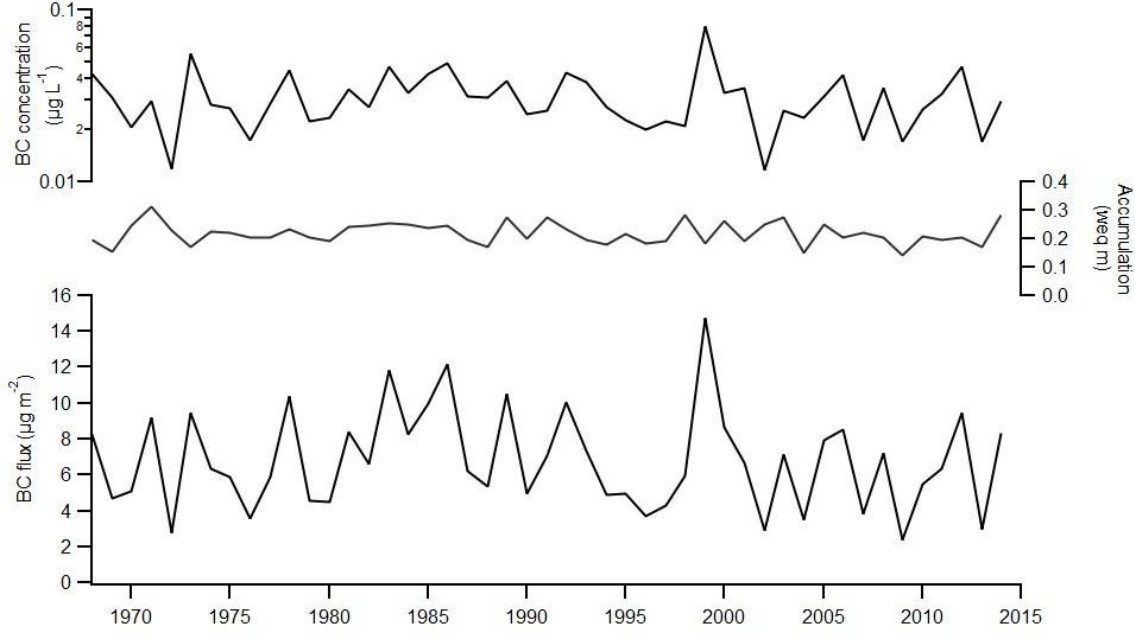

**Figure 5. rBC concentrations (y axis logarithmic), accumulation and fluxes for TT07.**

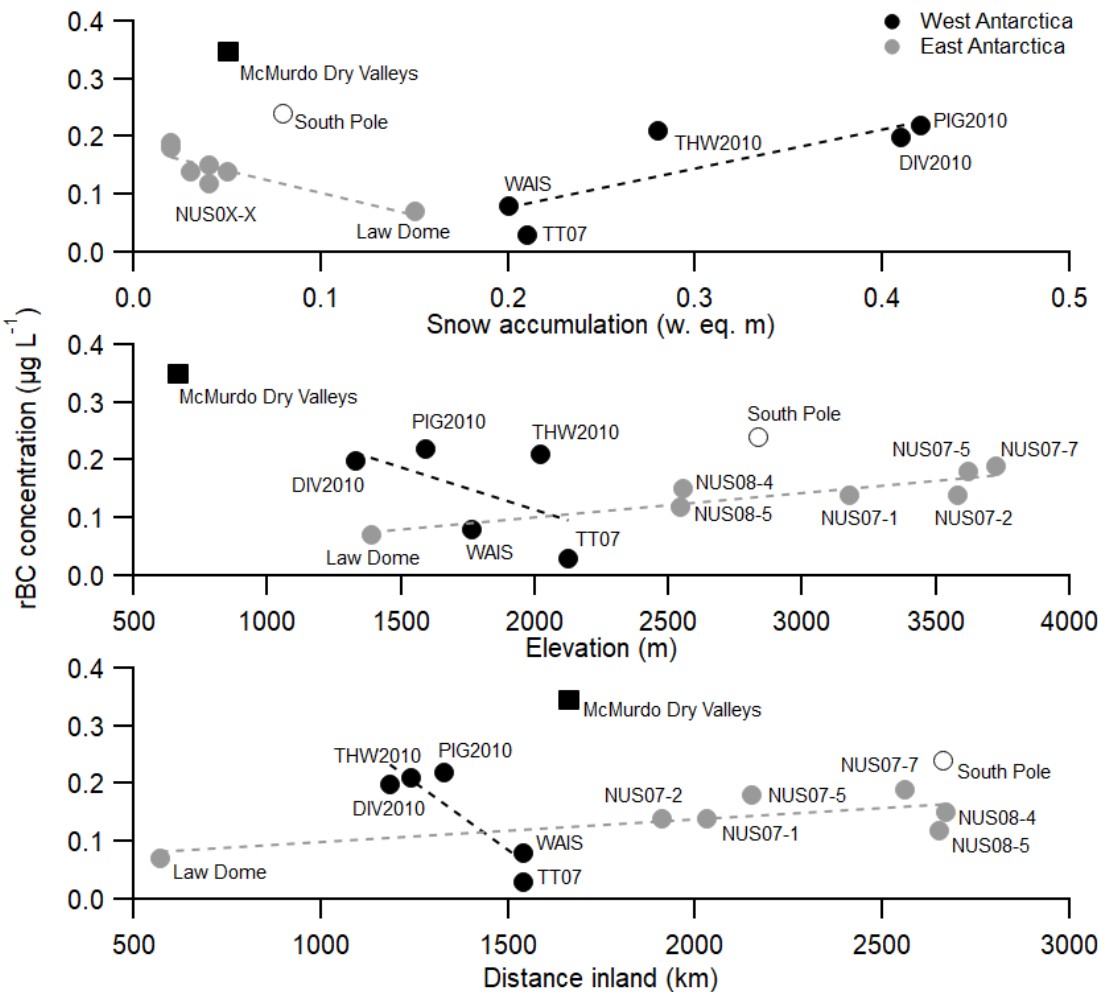

**Figure 6. rBC records from Antarctica. rBC concentrations plotted against snow accumulation, elevation and distance from the sea. Solid lines indicate statistically significant correlations (p < 0.05), while dashed lines indicate not significant correlations (p > 0.05).**

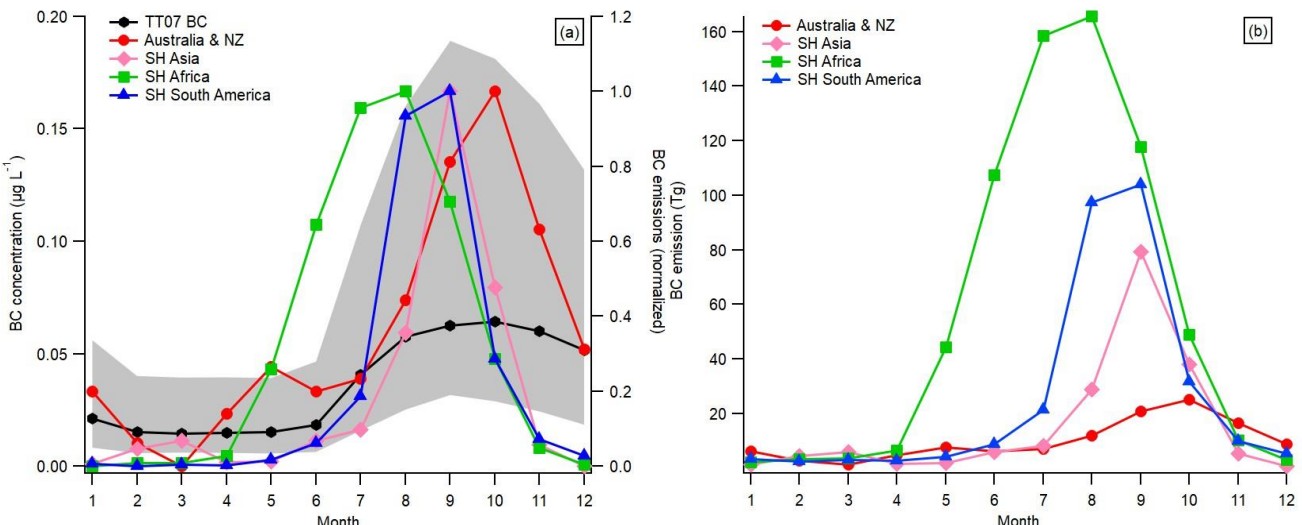

Figure 7(a). TT07 rBC (monthly averages, 1968-2014) and BC emissions estimated from GFED4s for the four SH regions (normalized, 1997-2014). The shaded area represent 1 geometric standard deviation of monthly rBC values. (b) Absolute BC emissions estimated from GFED4s for the SH.

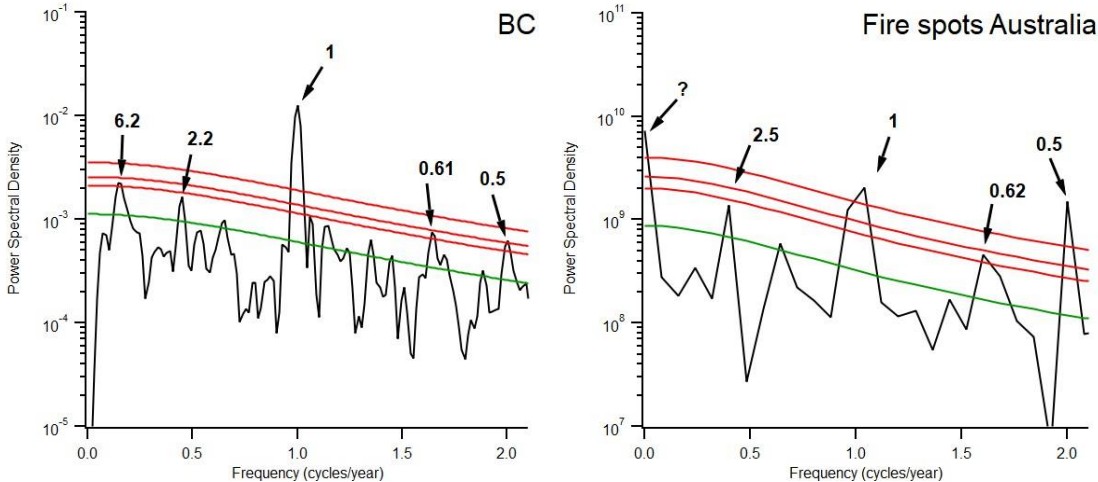

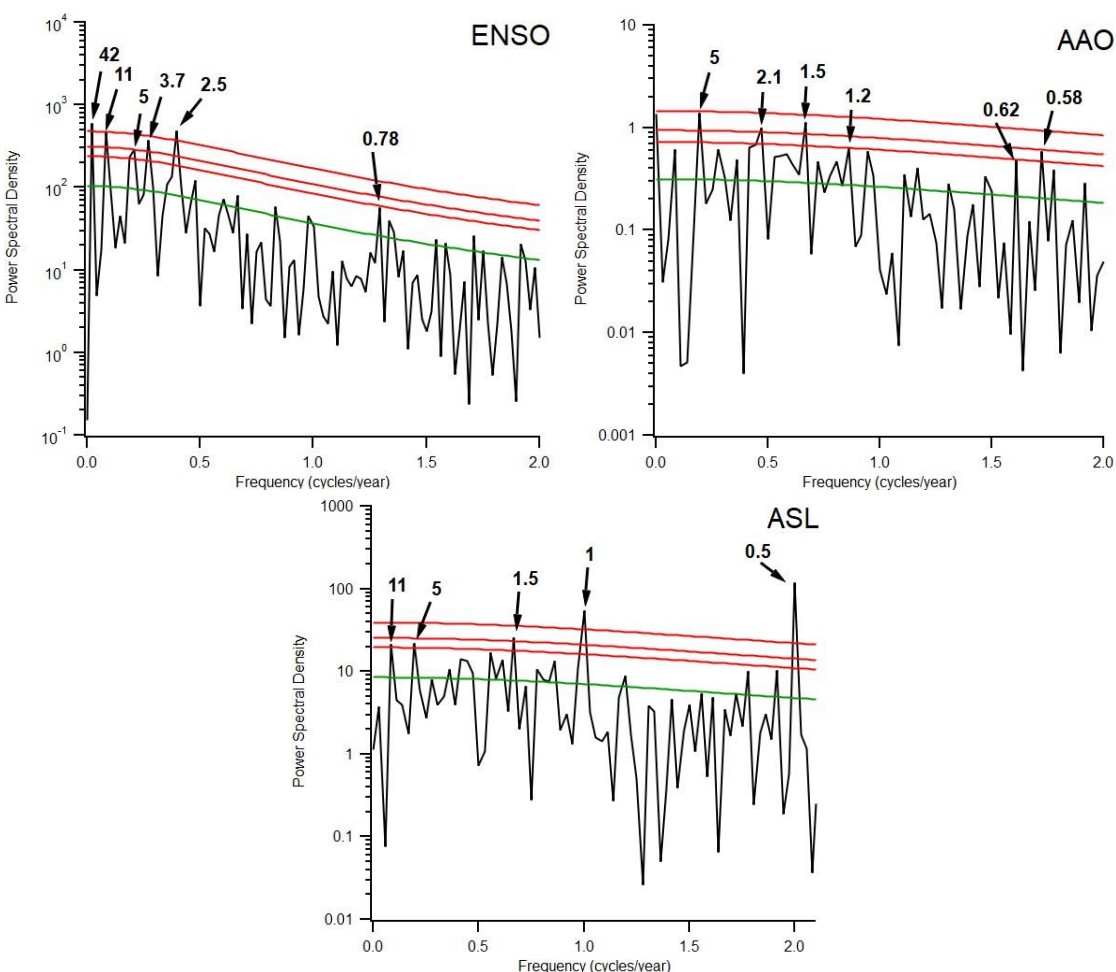

**Figure 8. Spectral analysis of the rBC concentrations and comparison with existing datasets (Sentinel Hotspots Australia, ASL, AAO and ENSO indexes). Numbers in bold indicate cycle frequency, in years. Red lines are confidence intervals 99% (top), 95% (middle) and 90% (bottom). Green line indicates AR1 red-noise background. The question mark in the Australian fire spots**
**spectrum indicates a longer, unidentified cycle.**

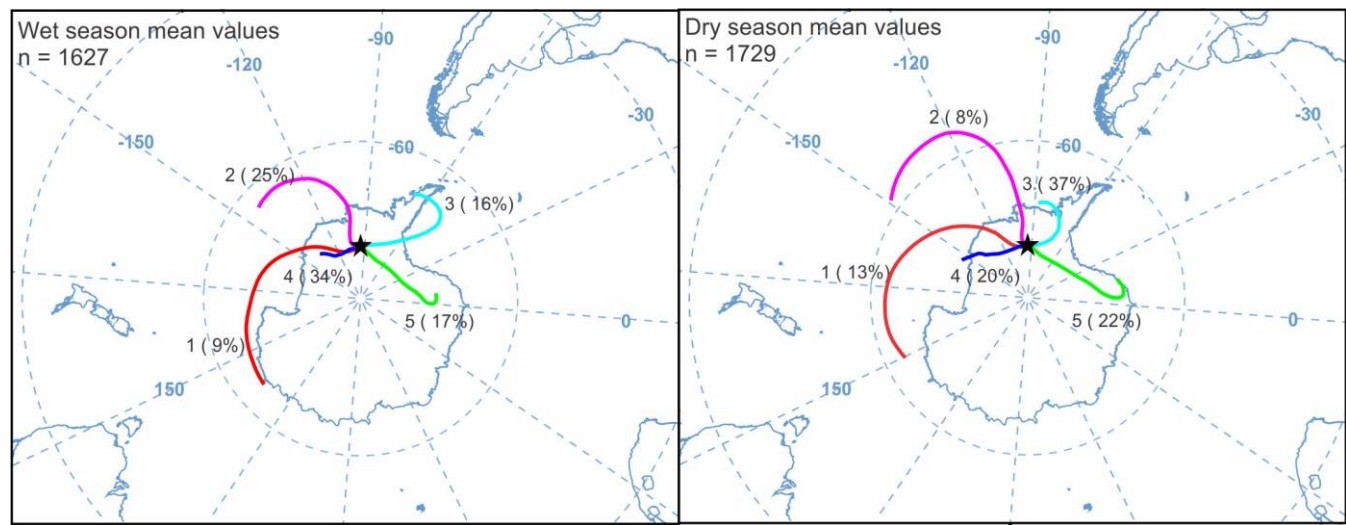

**Figure 9. HYSPLIT clusters of 10-day back-trajectories ran every 5 days from 1968 to 2015 arriving at the TT07 drilling site. Results are separated by wet and dry season, and grouped in five clusters (percentage of trajectories for each cluster is shown in parenthesis). Number of trajectories (n) used for the cluster algorithm is shown at the top, on the left side.**

| Wet Season | Dry Season | Wet Season | Dry Season |
|---|---|---|---|
| Cluster 1 of 5<br>n = 139 | Cluster 1 of 5<br>n = 226 | Cluster 4 of 5<br>n = 557 | Cluster 4 of 5<br>n = 352 |
| Cluster 2 of 5<br>n = 401 | Cluster 2 of 5<br>n = 142 | Cluster 5 of 5<br>n = 269 | Cluster 5 of 5<br>n = 376 |
| Cluster 3 of 5<br>n = 261 | Cluster 3 of 5<br>n = 633 | Ungrouped trajectories<br>n = 76 | |

**Figure 10. Individual trajectories used for the cluster analysis in figure 8. Number of trajectories (n) used for each cluster is shown at the top, on the left side. Clusters 1, 2 and 4 show air masses arriving from Australia and New Zealand to the TT07 drilling site,**

**while clusters 2, 3 and 4 show the (limited) contribution of South American air parcels to the site. Similar clusters from wet and dry season are side by side for comparison. Wet season presented 76 ungrouped trajectories, while dry season presented none.**

**Table 1. Parameters used to calculate albedo changes in snow for the TT07 site.**

| Incident-Flux | Diffuse |
|---|---|
| Surface spectral distribution | Summit Greenland clear-sky |
| Snowpack effective grain size | 150 μm |
| Snowpack thickness | 20 m |
| Snowpack density | 400 kg/m$^3$ |
| Visible albedo of underlying surface | 0.2 |
| Near-IR albedo of underlying surface | 0.4 |
| Uncoated black carbon concentration | Varied, see text |
| Sulfate-coated black carbon concentration | 0 ppb |
| Dust concentration | 0 ppm |
| Volcanic ash concentration | 0 ppm |
| Experimental particle 1 concentration | 0 ppb |
| MAC scaling factor | 1.0 |

**Table 2. Datasets used for the REDFIT spectral analysis.**

| Dataset | Data points | Range | Observation | Source |
|---|---|---|---|---|
| rBC | 860 | January 1969 December 2014 | raw data[a] | TT07 core |
| ENSO[b] | 576 | January 1967 December 2014 | Monthly data | Bureau of Meteorology, Australia[c] |
| AAO | 432 | January 1979 December 2014 | Monthly data | NOAA[d] |
| ASL | 432 | January 1979 December 2014 | Monthly data | BAS, Hosking *et al.* (2016)[e] |
| GFED4s | 216 | January 1997 December 2014 | Monthly data | Global Fire Emission Database |
| Sentinel Hotspots | 150 | August 2002 December 2014 | Monthly data | Geoscience Australia |

| Programa Queimadas | 200 | May 1998 December 2014 | Monthly data | INPE |

[a] Not resampled, only dated by year and separated by dry/wet season.
[b] Here we use the Southern Oscillation index – SOI as the ENSO indicator.
[c] http://www.bom.gov.au/climate/current/soihtm1.shtml
[d] https://www.cpc.ncep.noaa.gov/products/precip/CWlink/
[e] https://legacy.bas.ac.uk/data/absl/

**Table 3. Main results from the core rBC analysis. All values in µg L$^{-1}$, except fluxes, that are in µg m$^{-2}$ yr$^{-1}$. Geomean = geometric mean and 1σ\* = multiplicative standard deviation, representing 68.3% of the variability** (Bisiaux et al., 2012b; Limpert et al., 2001)**.**

| | |
|---|---|
| Total samples | 860 |
| Annual geomean | 0.03 |
| 1σ\* interval | 0.020 / 0.041 |
| Lowest/highest | 0.012 / 0.080 |
| Dry season geomean | 0.057 |
| 1σ\* interval | 0.031 to 0.105 |
| Lowest/highest | 0.005 / 0.332 |
| Wet season geomean | 0.015 |
| 1σ\* interval | 0.009 to 0.027 |
| Lowest/highest | 0.001 / 0.053 |
| rBC flux geomean | 6.25 |
| Lowest/highest | 2.67 / 14.61 |

**Table 4. Coordinates, elevation, period covered and rBC information for this study and previous studies in Antarctica with time overlap with this study. We show only studies that used the SP2 in snow/ice to have a direct comparison between them.**

| | | Location in Antarctica | Lat/ Long | Elev. (m) | Period Covered | Annual rBC conc. (µg L$^{-1}$) | rBC conc. range (2σ)[a] | Annual accum. (w eq m) | Annual rBC fluxes (µg m$^{-2}$) | rBC flux range (2σ)[a] |
|---|---|---|---|---|---|---|---|---|---|---|
| This study | TT07 | West | 82°40'S 89°55'W | 2122 | 1968-2015 | 0.03 | 0.01 to 0.06 | 0.21 | 6.25 | 2.7 to 14.6 |
| Pasteris et al. (2014) | DIV2010 | West | 76°48'S 101°42'W | 1329 | 1867-2010 | 0.17 | - | 0.41[b] | - | - |
| | PIG2010 | | 78°00'S 96°00'W | 1593 | 1917-2010 | | - | 0.42[b] | - | - |
| | THW2010 | | 76°46'S 121°13'W | 2020 | 1867-2010 | | - | 0.28[b] | - | - |
| Bisiaux et al., 2012b | WAIS | West | 79°46'S 112°08'W | 1766 | 1963-2001[c] | 0.08 | 0.05 to 0.12 | 0.2 | 16 | 9.8 to 24.4 |
| | Law Dome | East | 66°73'S 112°83'E | 1390 | 1963-2001[c] | 0.07 | 0.04 to 0.15 | 0.15 | 13.5 | 7.3 to 30.6 |
| Bisiaux et al., | NUS07-1 | East | 73°43'S 07°59'E | 3174 | 1963-2006[c] | 0.14 | 0.08 to 0.27 | 0.05 | 7.8 | 4.0 to 15.8 |

| | | | | | | | | | | |
|---|---|---|---|---|---|---|---|---|---|---|
| 2012b | NUS07-2 | | 76°04'S 22°28'E | 3582 | 1963-1993[c] | 0.14 | 0.08 to 0.24 | 0.03 | 3.9 | 2.4 to 7.3 |
| | NUS07-5 | | 78°39'S 35°38'E | 3619 | 1963-1989[c] | 0.18 | 0.14 to 0.24 | 0.02 | 3.62 | 2.6 to 5.2 |
| | NUS07-7 | | 82°49'S 54°53'E | 3725 | 1963-2008[c] | 0.19 | 0.13 to 0.29 | 0.02 | 5 | 3.1 to 8.0 |
| | NUS08-4 | | 82°49'S 18°54'E | 2552 | 1963-2004[c] | 0.15 | 0.08 to 0.26 | 0.04 | 5.4 | 2.7 to 10.0 |
| | NUS08-5 | | 82°38'S 17°52'E | 2544 | 1963-1993[c] | 0.12 | 0.08 to 0.20 | 0.04 | 4.5 | 2.7 to 8.0 |
| Casey et al., 2017 | Clean air sector | South Pole | 90°S | 2835 | Surface snow (2014-2015) | 0.24[d] | - | 0.08[e] | - | - |
| | Upwind of generator | | | | | 0.48[d] | - | | - | - |
| Khan et al., 2018 | Snowpit 1 m deep | McMurdo Dry Valleys | 77°31'S 163°E | 650 | 2006-2013 | 0.35[d] | - | 0.05[f] | - | - |

[a] Multiplicative standard deviation representing 95.5% of the interval of confidence.
[b] From Criscitiello et al. (2014).
[c] Core goes back to ~1800, we present only from 1963 on to have time overlap between these and this study.
[d] Not annual.
[e] From Mosley-Thompson et al. (1999)
[f] From Witherow and Lyons (2008)

**Table 5. Albedo changes due to rBC concentrations in TT07 site (from SNICAR-online).**

| Concentration (µg L$^{-1}$) | Reference | Albedo variation (relative to clean snow) |
|---|---|---|
| 0.015 | Wet season geomean | 0 |
| 0.057 | Dry season geomean | -0.41% |
| 0.105 | Highest seasonal geomean | -0.48% |
