# Peer review of "Refractory Black carbon (rBC) variability in a 47-year West Antarctic Snow and Firn core"

_The Cryosphere, 2019_

## Referee Comment (RC1) · Anonymous Referee #1 · 19 Dec 2019

This manuscript by Marquetto et al presents a 47 yr black carbon record from a 20-m firn core recovered from the Pine Island Glacier in West Antarctica. The authors measured rBC using the SP2 method, and dated the core primarily using the seasonal cycle of rBC. Potential southern hemisphere rBC source regions to this site were explored by correlating the seasonal cycle of rBC to Australian and South American fire spot data and GFED biomass burning emissions estimates as well as by comparing the power spectrum of rBC to large-scale atmospheric patterns (ENSO, AAO, and ASL) and fire spot data.

While Antarctic rBC records are important for understanding changes in southern hemisphere biomass burning as well as radiative forcing, and undoubtedly an immense amount of work went into developing this highly-resolved dataset, the interpretation

and discussion are not thorough or novel enough to add significantly to the understanding of rBC deposition in West Antarctica. The discussion focused on three analyses: SNICAR snow albedo modeling, identifying continental emissions sources, and linking rBC to atmospheric circulation using spectral analysis. The SNICAR modeling showed that rBC deposition at this site has little to no effect on snow albedo, which is not unexpected given the extremely low rBC concentrations and (as noted by the authors) has already been shown for clean Antarctic snow (Casey et al., 2017). The identification of source regions by comparing seasonal cycles in observed rBC and GFED fire emissions is purely correlative—I do not think is a strong enough approach, especially with my concerns about dating, to draw conclusions about source regions without a more robust approach that would consider atmospheric transport and magnitude of biomass burning emissions. Finally, the spectral analysis was similar to that conducted by Bisiaux et al. (2012) for the WAIS Divide and Law Dome rBC records and does not provide any concrete new links between atmospheric circulation and Antarctic rBC. Furthermore, I have concerns about the factor of 2-3 lower rBC concentrations compared to other West Antarctic ice cores (see comments below).

Overall, since the conclusions do not add substantial new insight or understanding to rBC deposition and mainly confirm what is already known, I do not think this manuscript is suitable for the scope of The Cryosphere.

I have two major concerns about the methods: 1.) rBC measurements, and 2.) dating. 1. rBC measurements: The rBC concentrations presented in this study are a factor of 2-3 lower than rBC concentrations in other West Antarctic cores, including the WAIS Divide ice core as well as other measurements from Pine Island Glacier (Pasteris et al., 2014). Note that rBC measurements from early 1900s to 2006 from Pine Island Glacier (as well as Thwaites Glacier and the divide between Pine Island and Thwaites Glaciers) were previously published in Pasteris et al. (2014) and are also 2-3x higher than the concentrations presented in this manuscript- it would be worth including this citation and even comparing to this published dataset to see how the magnitude and temporal

variability of the records compare. The authors need to provide more information to determine if this offset is real or a result of the SP2 calibration/analytical system. Since the referenced Marquetto et al. (2019) manuscript does not appear to be published at the time of this review, please include details on the SP2 internal/external calibration in this manuscript. How were the ice samples melted (room temperature?), and how soon before SP2 analysis were they melted (lines 96-97)? Wendl et al. (2014) show how rBC is lost after melting during sample storage in polypropylene vials over just a few days (with proportionally greater losses for low-concentration samples). How often was 5% $HNO_3$ used to clean the system (line 118), and how did you determine when the acid was flushed from the system? Wendl et al. (2014) also discuss how acidification can result in significant loss of rBC. What kind of tubing was used to transport the aerosol from the Marin 5 nebulizer to the SP2?

2. The description of the dating, namely the role of the datasets from the Schwanck et al. (2017) study, is vague. Since much of the analysis, including the seasonal cycle correlations and spectral analysis, require precise dating, more explanation of the dating must be given. The annual picks in Fig. 1 are consistently on the austral summer (January) decrease of rBC concentration, but appear to be inconsistently placed across the S, Sr, and Na records. Is the S, Sr, and Na data shown in Fig. 1 all from the Schwanck et al. core from ~1 km away? I would expect cores 1 km apart to have a depth offset, so I would not be confident in correlating the picks from TT07 to the Schwanck et al. core without a common dataset to both cores to linking the cores in depth. Without showing that the chemical species from the TT07 core and Schwanck core are not offset in depth, it is not advisable to guide the dating of the TT07 core with data from the Schwanck et al. core, and likewise not justified to apply annual picks based on rBC in TT07 core to the Schwanck core Na record (I can't tell if the Na spectral analysis was conducted on the TT07 age scale or the original Schwanck et al. age scale). How does the Schwanck et al. data compare to the 5.6 m of S, Sr, and Na data for the core taken immediately adjacent to the rBC core (mentioned on lines 154-155)? There should be a few years of overlapping data to compare and that would

at least be a start to justify comparing the two cores in depth.

Furthermore, there appears to be circular logic between the dating and seasonal comparison to GFED. If indeed the drop in rBC concentration at the end of the austral summer was used as the primary annual pick, and this pick was justified by GFED/fire spot seasonality as stated in lines 161-163, how can you then draw meaningful conclusions about the timing of rBC vs. GFED (in Fig. 6a) to identify source regions? The rBC and GFED are already inherently linked based on how you defined the dating of the core. Based on GFED seasonality in Fig. 6a, BC emissions drop in November for Asia/Africa/S. America, two months before the January drop for Australia/NZ. Even a month or two difference on where the rBC drop is assigned could have significant implications for the correlations used in section 4.6 (based on n=12 months) which are used to underpin the conclusion on lines 286-289. It would be much more appropriate to date the TT07 core using an independent chemical species, and then use the independent dating to examine the rBC seasonality.

Other comments- Lines 14-15: Please specify what you mean by wet and dry season? I assume southern hemisphere wet/dry season, not at the ice core site (also on lines 145-146).

Lines 87 and 82: Is Marquetto et al. (2019) published and available?

Lines 83 and 90: Same heading title for sections 3.2 and 3.3.

Line 116: Did you average the rBC data for the full 5 minutes that it was run? Can you quantify the stability of the measurement with a standard deviation over the time period averaged?

Lines 135-143: The Brazilian hotspot data is defined here, but never mentioned in the results and discussion section. Can it be omitted? Or did it result in a null finding?

Line 140: How do the timeseries of the fire hotspot data compare to the rBC data? You only compare the power spectra in this study. Do years with more hotspots

correspond to years with more rBC deposition?

Lines 147-148: Please define Na, Sr, and S before using abbreviations

Lines 167-173: Please plot the average of the two density profiles in Fig. 3 that the quadratic equation was fit to.

Line 180: Please explicitly state which months went into the summer/fall and winter/spring averages.

Line 193 and 296: Is the Na record you use for spectral analysis from the TT07 core, or is it from Schwanck et al.? If it is from Schwanck et al., have you changed the dating?

Lines 211-212: Per comments above, please clarify what dating is used for which chemical species.

Lines 244-246: Please include comparison to rBC data from Pasteris et al. (2014) from Pine Island Glacier.

Line 338: Abstract says record extends from 1968-2015.

Figure 1: Please check location of B40 ice core site.

Figure 2: Please define which records are from the TT07 core and which are from Schwank et al.

Figures 4, 5: It would be helpful to include the standard deviation (or standard error) of the measurements that went into the annual averages and wet/dry season plots as an error bar.

Figures 6, 8: Again, it would be helpful to have an estimate of uncertainty on the seasonal cycles.

Figure 7: Did you use rBC flux or concentration for the spectral analysis?

Figure 8: Could the broad Na peak over 4 months be a result of mixing/matching Na data and depth picks between the TT07 and Schwanck et al. cores?

Table 4: Please include Pasteris et al. (2014) rBC data.

References: Pasteris, D. R., J. R. McConnell, S. B. Das, A. S. Criscitiello, M. J. Evans, O. J. Maselli, M. Sigl, and L. Layman (2014), Seasonally resolved ice core records from West Antarctica indicate a sea ice source of sea-salt aerosol and a biomass burning source of ammonium, J. Geophys. Res.Atmos.,119, 9168–9182, doi:10.1002/2013JD020720

Wendl, I. A., Menking, J. A., Färber, R., Gysel, M., Kaspari, S. D., Laborde, M. J. G., and Schwikowski, M.: Optimized method for black carbon analysis in ice and snow using the Single Particle Soot Photometer, Atmos. Meas. Tech., 7, 2667–2681, https://doi.org/10.5194/amt-7-2667-2014, 2014.

---

## Referee Comment (RC2) · Anonymous Referee #2 · 1 Jan 2020

The paper provides a 47-year ice core record of refractory black carbon (rBC) from West Antarctica, specifically Pine Island Glacier. rBC was analyzed by a Single Particle Soot Photometer. The core was dated to 1968, primarily using seasonality of rBC. BC impacts on snow albedo were modeled using the Snow, Ice, Aerosol, Radiation (SNICAR) model. BC emissions were explored with fire spot inventories and spectral analysis was conducted by the REDFIT method.

With respect to the TC guidelines: 1. The paper provides additional field observations of rBC in snow and ice in a data-sparse region of the cryosphere. Making field observations like this available to the community is important for refining our understanding of impurities in the cryosphere and their impact on surface albedo of the Antarctic ice sheet. 2. While the paper provides an additional valuable dataset, it is unclear to me

whether the record interpretation is particularly novel. 3. The main finding appears to be that BC transport to the site is not related to marine air masses, which has previously been shown in other ice core records in Antarctica (i.e. Bisiaux et al., 2012). 4. The analytical details appear to be outline in Marquetto et al., 2019, however, I am having trouble locating the manuscript. 5. Thus, I have some remaining analytical questions outlined below. 6. Having access to Marquetto et al., 2019 would assist with reproducibility. 7. The authors provide credit to related work, but I think they should further identify/emphasize the novelty of their contribution. 8. The title clearly reflects the content of the paper. 9. The abstract provides a concise and complete summary of the existing manuscript. 10. The paper could have benefited from more thorough proofreading before submission; there are some typos. 11. Please refer to 10. 12. Black carbon and refractory black carbon are abbreviated at times and then spelled out at others (i.e. Lines 36, 67, 338). 13. Current figures and tables seem to appropriately support the text. 14. The number of references seems appropriate. 15. The Marquetto et al., 2019 paper would have been useful supplementary information.

Specific Suggestions:

Line 29: Typo: 'while there they change'

Line 45 – 48: Sentence could be restructured for clarity.

Line 58 – 61: This section could be expanded, including more specific references.

Line 77: I cannot find Marquetto et al., 2019 online. Thus, a lot of important analytical details seem to be missing from this manuscript. For example, how long before analysis were the samples melted?

Lines 83 and 90: Duplicate sub-section titles.

Line 87: Why were polypropylene vials used instead of glass vials? Was particle loss explored with leaching on the vials? Additionally, how long did the samples sit in the vials before analysis?

Line 146: I don't think Sr is mentioned in Legand and Mayewski, 1997.

Line 197: Suggest 'fit' as opposed to 'fitted'.

Line 199: Suggest using the same past tense, 'chose' as opposed to 'choose'.

Section 4.1 Dating: Given that the main findings of the paper rely on dating based on seasonality of the rBC record, I think this section could be expanded. For example, the authors could add more discussion as to why the authors think the addition of the rBC record to the layer counting would lead to a dating difference of one year or more, with respect to the core collected nearby that was analyzed for trace elements.

Line 338: The starting date here (1969 – 2015) does not match the abstract or Table 4 (1968 – 2015).

Figure 4 Legend: Suggest (bottom) instead of (base).

---

## Author Comment (AC2) · 27 Feb 2020

We apologize for the inconvenience that Marquetto et al. (2019), cited in this manuscript, was not available during the review process. Marquetto et al. (2019) was accepted for publication in October 2019 and will be available online in April 2020 in "Advances in Atmospheric Sciences" as Marquetto et al. (2020). Please find the accepted manuscript as supplementary material in this response.

Please also note the supplement to this comment:
https://www.the-cryosphere-discuss.net/tc-2019-207/tc-2019-207-AC2-supplement.pdf

---

## Author Comment (AC4) · 27 Feb 2020

| 1  | BC RECORD IN A FIRN CORE FROM WEST ANTARCTICA                                                                    |  |  |  |  |  |
|----|------------------------------------------------------------------------------------------------------------------|--|--|--|--|--|
| 2  | MARQUETTO ET AL.                                                                                                 |  |  |  |  |  |
| 3  | VOL. 37, APRIL 2020, 1-10                                                                                        |  |  |  |  |  |
| 4  |                                                                                                                  |  |  |  |  |  |
| 5  | Refractory Black Carbon Results and a Method Comparison between Solid-state                                      |  |  |  |  |  |
| 6  | Cutting and Continuous Melting Sampling of a West Antarctic Snow and Firn Core                                   |  |  |  |  |  |
| 7  | Luciano MARQUETTO *1,2 , Susan KASPARI 1 , Jefferson Cardia SIMÕES 2 , and Emil |  |  |  |  |  |
| 8  | $BABIK^1$                                                                                                        |  |  |  |  |  |
| 9  | 1 Department of Geological Sciences, Central Washington University, Ellensburg,                       |  |  |  |  |  |
| 10 | Washington 98926, USA                                                                                            |  |  |  |  |  |
| 11 | 2 Polar and Climatic Center, Federal University of Rio Grande do Sul, Porto Alegre, Rio               |  |  |  |  |  |
| 12 | Grande do Sul, 91509-900, Brazil                                                                                 |  |  |  |  |  |
| 13 | (Received 12 July 2019; revised 4 September 2019; accepted 24 October 2019)                                      |  |  |  |  |  |
| 14 | ABSTRACT                                                                                                         |  |  |  |  |  |
| 15 | This work presents the refractory black carbon (rBC) results of a snow and firn core drilled                     |  |  |  |  |  |
| 16 | in West Antarctica (79°55'34.6"S, 94°21'13.3"W) during the 201415 austral summer,                                |  |  |  |  |  |
| 17 | collected by Brazilian researchers as part of the First Brazilian West Antarctic Ice Sheet                       |  |  |  |  |  |
| 18 | Traverse. The core was drilled to a depth of 20 m, and we present the results of the first 8 m by                |  |  |  |  |  |
| 19 | comparing two subsampling methodssolid-state cutting and continuous meltingboth with                             |  |  |  |  |  |
| 20 | discrete sampling. The core was analyzed at the Department of Geological Sciences, Central                       |  |  |  |  |  |
| 21 | Washington University (CWU), WA, USA, using a single particle soot photometer (SP2)                              |  |  |  |  |  |
| 22 | coupled to a CETAC Marin-5 nebulizer. The continuous melting system was recently                                 |  |  |  |  |  |

\* Corresponding author: Luciano MARQUETTO E-mail: luciano.marquetto@gmail.com

23 assembled at CWU and these are its first results. We also present experimental results regarding SP2 reproducibility, indicating that sample concentration has a greater influence than the 24 analysis time on the reproducibility for low rBC concentrations, like those found in the 25 26 Antarctic core. Dating was carried out using mainly the rBC variation and sulfur, sodium and strontium as secondary parameters, giving the core 17 years (1998--2014). The data show a 27 28 well-defined seasonality of rBC concentrations for these first meters, with geometric mean summer/fall concentrations of 0.016  $\mu$ g L-1 and geometric mean winter/spring concentrations 29 of 0.063  $\mu$ g L-1. The annual rBC concentration geometric mean was 0.029  $\mu$ g L-1 (the lowest 30 31 of all rBC cores in Antarctica referenced in this work), while the annual rBC flux was 6.1 µg  $m^{-2}$  yr-1 (the lowest flux in West Antarctica records so far). 32

33 Key words: black carbon, West Antarctica, ice core; single particle soot photometer

Citation: Marquetto, L., S. Kaspari, J. C. Simões, and E. Babik, 2020: Refractory black carbon
results and a method comparison between solid-state cutting and continuous melting sampling
of a West Antarctic snow and firn core. *Adv. Atmos. Sci.*, 37(4), 000--000,
https://doi.org/10.1007/s00376-019-9124-8.

**38** Article Highlights:**

- The continuous melting system with discrete sampling is a faster and reliable way of
   analyzing low rBC concentration samples.
- The record showed a well-defined seasonal signal for black carbon, with higher
   concentrations during the Southern Hemisphere dry season.
- The sample concentration influences the analysis reproducibility more than the analysis
   time.
- The annual black carbon concentration was lower than other West Antarctic records and
   comparable to high-elevation East Antarctica ice cores.

**47 **1. Introduction**

Black carbon (BC) is a carbonaceous aerosol formed during the incomplete combustion of biomass and fossil fuels, characterized by strong absorption of visible light and resistance to chemical transformation (Petzold et al., 2013). Increases in BC concentrations since the industrial revolution have been observed in ice sheets and caps around the world, with direct implications for the planetary albedo (Hansen and Nazarenko, 2004; Bice et al., 2009; Bond et al., 2013).

54 Approximately 80% of Southern Hemisphere BC emissions are from in-situ biomass 55 burning, mainly from forest and savannah deforestation (Bice et al., 2009), with 80%--95% of 56 this burning being human-related (Lauk and Erb, 2009). Ice cores retrieved from the Antarctic 57 continent record these Southern Hemisphere emissions and long-range transport of BC from 58 low- and midlatitudes (Bisiaux et al., 2012a, b). Long-range transport of BC from low- and 59 midlatitudes to the polar ice caps is possible due to BC's insolubility, graphite-like structure 60 and small size (< 10 nm to 50 nm diameter), resulting in low chemical reactivity in the 61 atmosphere and slow removal by clouds and precipitation unless coated with water-soluble compounds (Petzold et al., 2013). BC concentrations in Antarctica have already been linked to 62 63 biomass burning from South America, Africa and Australia (Koch et al., 2007; Stohl and Sodemann, 2010; Arienzo et al., 2017). Although there are records of Southern Hemisphere 64 65 paleo-biomass burning (Marlon et al., 2008, 2016; Wang et al., 2010; Osmont et al., 2018a), 66 there are only a few recent publications on BC variability in ice cores from Antarctica (Bisiaux et al., 2012a, b; Arienzo et al., 2017). More ice core records from different time scales are 67 needed to understand the spatial variability of BC transport to, and deposition in, Antarctica, 68 69 as well as to improve general circulation models (Bisiaux et al., 2012b).

This work discusses two subsampling methods [solid-state cutting (SSC) and a continuous
melting system (CMS)], and the system setup, used to analyze refractory black carbon (rBC)

in the first eight sections of a snow and firn core collected in West Antarctica. A preliminary
 environmental interpretation is also presented.

**74 **2. Site description and field campaign**

75 The core was drilled on the Pine Island Glacier at (79°55'34.6"S, 94°21'13.3"W; elevation: 76 2122 MSL), near the Mount Johns Nunatak (located 70 km northeast of the drilling site) where 77 the ice thickness is  $2400 \pm 300$  m (Fretwell et al., 2013) (Fig. 1). The majority of air masses 78 arrive from the Amundsen Sea and, secondarily, from across the Antarctic Peninsula and 79 Weddell Sea (Schwanck et al., 2017). As stated by Schwanck et al. (2016b), the site was chosen 80 due to (1) its relatively high accumulation rate (0.21 water equivalent meters per year); (2) it is 81 a drainage basin divide (between the Pine Island and Institute Glacier); and (3) it is an area 82 where air masses from the Weddell, Amundsen and Bellingshausen seas converge. It is located 83 approximately 350 km from the West Antarctic Ice Sheet (WAIS) Divide drilling site, from 84 where Bisiaux et al. (2012b) recovered the first, and until now, only high-temporal-resolution 85 rBC record from West Antarctica covering recent decades.

The drilling was part of the First Brazilian West Antarctic Ice Sheet Traverse, carried out 86 87 in the austral summer of 2014--15 along a 1440-km route from Union Glacier (79°46'05"S 88 83°15'42"W) in the Ellsworth Mountains, to the automated Brazilian atmospheric module Criosfera 1 (84°00'S, 79°30'W), and then to the Mount Johns area (79°55'34.6"S, 89 90 94°21'13.3"W). We used a Mark III auger (Kovacs Enterprises, Inc., Roseburg, OR, USA) 91 coupled with an electrical drive powered by a generator (kept downwind at a minimum of 30 92 m away) to retrieve all cores in the traverse. The core presented in this study (TT07) was drilled 93 from the surface to a total depth of 20.16 m, divided in 21 sections of less than 1 m each. The 94 borehole temperature was  $-34^{\circ}$ C, measured at 12 m deep by a probe previously calibrated that 95 remained in the borehole for at least 8 h.

All sections of the core were weighed in the field, packed in polyethylene bags and then stored in high-density styrofoam boxes. These boxes were sent by air to Punta Arenas (Chile), then to a deposit in Bangor (USA) for storage, and finally to the Central Washington University (CWU) Ice Core Laboratory (Ellensburg, WA), where it was kept at -18°C in a clean, cold room until subsampling and analysis.

**101 **3. Materials and methods**

**102 **3.1** *rBC* analysis in snow and ice samples**

We used an extended range single particle soot photometer (SP2, Droplet Measurement Technologies, Boulder, CO, USA) to analyze the core, and thereby our results are measurements of rBC (Petzold et al., 2013). The particle size range detected by the SP2 at CWU was 80--2000 nm (mass-equivalent diameter) for the incandescent signal, assuming a void-free BC density of 1.8 g cm-3 (Moteki and Kondo, 2010).

The SP2 was initially designed to measure rBC in the atmosphere, and then adapted to analyze snow and ice samples. It was first used by McConnell et al. (2007) to analyze an ice core retrieved from Greenland spanning 1788--2000 AD, and since then the method has been applied in numerous studies (Kaspari et al., 2011; Bisiaux et al., 2012a, b; Kaspari et al., 2014, 2015; Casey et al., 2017; Osmont et al., 2018a, b; Sigl et al., 2018). As the system was designed to analyze airborne samples, it is necessary to add an aerosolization step in order to analyze the snow and ice meltwater (Wendl et al., 2014).

For the SP2 external calibration (Wendl et al., 2014), five fresh standards ranging from 0.01 to 1.0  $\mu$ g L-1 were prepared every day in glass jars, by diluting a 4585.6  $\mu$ g L-1 Aquadag stock in Milli-Q water (MilliQ-Element, Millipore, Milford, USA – 18,2 M  $\Omega$  cm) previously sonicated for 15 min. Aquadag (Acheson Industries Inc., Port Huron, MI, USA) is an industrial, graphite-based lubricant consisting of a colloidal suspension of aggregates of graphitic carbon in water, with a content of BC between 71% and 76% of solid mass, proven suitable forcalibration standards by Wendl et al. (2014).

122 An environmental standard (diluted meltwater of a snow sample from Table Mountain, 123 WY, USA) of known concentration  $(0.18 \pm 0.04 \ \mu g \ L^{-1})$  was also analyzed every day, to ensure 124 there were no mistakes when preparing the Aquadag standards. The Aquadag stock and the 125 environmental standard were kept in closed glass jars and refrigerated at ~5°C when not in use, 126 and sonicated for 15 min prior to usage. For the nebulization step we used a CETAC Marin-5, 127 described by Mori et al. (2016).

128 Internal calibration of the SP2 (Wendl et al., 2014) was carried out using a known 129 polydisperse BC standard of aqueous Aquadag diluted in Milli-Q water. The Aquadag solution 130 was nebulized, and then passed through a sillica diffusion drier (to remove moisture) and an x-131 ray source (Advanced Aerosol Neutralizer Model 3088, TSI Inc., MN, USA) to neutralize 132 particle charges before entering a centrifugal particle mass analyzer (CPMA), similar to the set 133 up in Olfert et al. (2007) but without the differential mobility analyzer. The CPMA was 134 configured to select 23 particle masses from 0.5 fg to 800 fg. Each selected mass ran for 30 135 min to 6 h to provide statistically significant particle triggers to calibrate the SP2, and 136 calibration curves were then generated for all SP2 channels. The data presented here are from 137 the duplicated extended range broadband detector, as this channel gave the best-fit calibration 138 curve of all channels, with a precise fitting in the lower end of the particle mass range. For 139 more details on the calibration, see Table S1 and Fig. S1 in the electronic supplementary 140 material (ESM).

**141 **3.2** Sample preparation**

142 The sample preparation process consisted of removing the outer layers of the core, as these 143 are prone to contamination during drilling, handling and transport of the core (Tao et al., 2001). 144 Antarctic samples are especially sensitive to contamination owing to the very low 145 concentrations of analytes commonly observed in them. Previous works have shown rBC concentrations in West Antarctic snow to be as low as 0.01  $\mu$ g L-1 (Bisiaux et al., 2012b). 146 147 intensive cleaning was carried out inside the cold room for all Regular, 148 surfaces/parts/equipment in contact with the core using ethanol and laboratory-grade paper 149 tissues. Tyvek suits (DuPont, Wilmington, DE, USA) and sterile plastic gloves were used at 150 all times in the cold room during the core processing. Vials used to store the samples (50-mL polypropylene vials) were soaked in Milli-Q water for 24 h and rinsed three times. This process 151 152 was repeated two more times, in a total of three days soaked in Milli-Q water and nine rinses. 153 The vials were left to dry, covered from direct contact, in the laboratory.

154 We used two different methods to analyze the core in order to compare them: SSC and a 155 CMS. We partitioned the 21 sections of the core longitudinally, using a bandsaw with a meat 156 grade, stainless steel bandsaw blade, and samples from the same depths were prepared using 157 SSC and the CMS (the cut plan is presented in Fig. S2 in ESM). For every cutting session, a 158 Milli-Q ice stick, previously prepared, was cut in the beginning, to guarantee a clean blade for 159 the snow and firn core. For both methods, we hand-scraped the resulting snow and firn sticks 160 with a clean ceramic knife, to remove the outer snow/firn layer (2--4 mm). This process was 161 carried out in a laminar flow hood, still in the cold room.

162 3.2.1 *SSC*

163 The SSC method consisted of cutting the hand-scraped snow and firn sticks in 2--2.5-cm 164 samples with a ceramic knife, resulting in ~40 samples (of 6--8 mL each) per section. This 165 process was also carried out in the laminar flow hood. We stored the samples in pre-cleaned 166 50-mL polypropylene vials and kept them frozen until analysis.

Samples were melted at room temperature or in a tepid bath not exceeding 25°C, sonicated
for 15 min, and then analyzed (in less than 1 h after melting).

169 3.2.2 *The CMS*

We assembled a CMS at the CWU Ice Core Laboratory, based on the system developed at the Climate Change Institute (CCI), University of Maine, USA---described in detail in Osterberg et al. (2006) and used by Schwanck et al. (2016a, b, 2017). The main advantage of the CMS compared to SSC is the reduced handling of samples.

The inner part of the core was collected with a fraction collector in pre-cleaned 50-mL polypropylene vials for rBC analysis, resulting in ~43 samples (of 6--10 mL each) per section. The outer part of the core was discarded. The samples were kept refrigerated at 5°C until the time of analysis, and were then sonicated for 15 min and analyzed (less than 2 h after melting). As the flow remained constant, the sample depth was calculated by dividing the length of each section by the number of resulting samples.

180 The main differences between the melting system used at CCI and the one assembled at181 CWU are:

(1) The system was built to only collect samples for rBC, meaning we only collect the
melting water from the inside ring of the melting head. This also means we only use two
peristaltic pumps---one for the inside ring and another for the outside ring (wastewater).

(2) The melting disk at CWU is made of aluminum (not nickel); as we are not analyzing
samples for heavy metals, there is no need for a high-purity nickel disk.

(3) As samples are less prone to BC contamination in comparison to trace-element
contamination, the fraction collector linked to the melting system sits on a normal lab bench,
not in a flow hood.

(4) The melting head temperature during use is set to 10°C--15°C (instead of 15°C--20°C
as commonly used at CCI). Higher temperatures generate persistent wicking processes, and
this lower temperature range causes less of a problem (although wicking never stops)

completely). Due to time constraints related to the assembling and testing of the continuousmelter, we could only prepare and analyze eight sections with this method.

**195 **3.3** Whole-system setup**

196 After melting, the sample is dispensed to the Marin-5 nebulizer by a Regro Digital peristaltic pump (ISMATEC, Wertheim, Germany) at  $0.14 \pm 0.02$  mL min-1 and monitored by 197 a TruFlo Sample Monitor (Glass Expansion, Port Melbourne, Australia). The Marin-5 198 nebulizer receives standard laboratory air at 1000 sccm (1.000 L min-1), regulated by an Alicat 199 200 Flow Controller (Alicat Scientific, Tucson, AZ, USA) connected to a Drierite Gas Purifier, 201 which removes any moisture or particulates from the air. The nebulizer heating and cooling 202 temperatures are set to 110°C and 5°C, respectively, following Mori et al. (2016). We used Tygon Long Flex Life (LFL) tubing ID 1.02 mm (Saint-Gobain Performance Plastics, France) 203 204 for sample to nebulizer connection.

The SP2 flow was maintained at 120 volumetric  $cm^3 min^{-1}$  (vccm). YAG laser power for this project stayed constant above 5.0 V.

207 Procedural blanks (MQ water) were run at the beginning and end of every working day, 208 and also every 15--20 samples. Background levels were kept at 0--0.5 particles cm-3 and a 5% 209 nitric acid solution was used for cleaning the tubing and nebulizer when needed. For the SP2 210 to return to background levels, only MQ water was used. Peristaltic pump tubing replacement 211 was necessary only once during the process.

The limit of detection (LOD) of the method was estimated to be  $1.61 \times 10^{-3} \,\mu g \, L^{-1}$ , based on procedural blanks measured to characterize the instrument detection limit (mean +  $3\sigma$ , n =214 30).

Samples were analyzed for 5 min each, with a whole-system reproducibility test carried
out to assess the uncertainty related to the method. This test is presented in section 4.4.

Data processing was performed with the SP2 Toolkit 4.200 developed by the Laboratory
of Atmospheric Chemistry at the Paul Scherer Institute, and was used on the IGOR Pro version
6.3 scientific data analysis software.

**220 **3.4** *Fire-spots database**

To help define the dating of the core, we compared our rBC results with fire spots (number of active fires) detected by satellites for the Sentinel Hotspots program (Geoscience Australia, Australia, available at https://sentinel.ga.gov.au/) and Programa Queimadas (Instituto Nacional de Pesquisas Espaciais, Brazil, available at http://www.inpe.br/queimadas/portal). Both systems use the MODIS, AVHRR and VIIRS sensors to pinpoint fire spots. Sentinel Hotspots presents data from 2002 to present and covers Australia and New Zealand, while Programa Queimadas has data from 1998 to present, and covers all South American countries.

Although Africa has the highest total BC emissions of the Southern Hemisphere, the continent contributes little to BC in Antarctica because emissions are located further north than South American and Australian emissions (Stohl and Sodemann, 2010).

Even though the parameter "fire spot" used in both Australian and Brazilian fire monitoring programs does not translate directly to the dimension and intensity of the biomass burning events, it holds a correlation with burned area (Andela et al., 2017), and so we consider it useful to our comparison.

235 **3.5** *rBC* concentrations and fluxes

The frequency distributions of the TT07 core rBC concentrations were determined to be lognormal, and so we present geometric means and geometric standard deviations because these are more appropriate than arithmetic calculations (Limpert et al., 2001; Bisiaux et al., 2012a). Note that the geometric standard deviation is the multiplicative standard deviation ( $\sigma^*$ ), so the 68.3% confidence interval is calculated as  $\sigma min_{conc}$  = geometric mean × geometric standard deviation, and  $_{\sigma}$ maxconc = geometric mean / geometric standard deviation (Limpert et al., 2001).

We present our data as summer/fall (dry season) concentrations and winter/spring (wet season) concentrations. Wet/dry season concentrations and annual concentration geometric means and standard deviations were calculated in the raw rBC measurements using the dating carried out to separate years and rBC concentration variations to pinpoint the changes from dry season to wet season, and vice versa. Monthly mean concentrations were calculated by applying a linear interpolation in the raw measurements.

rBC fluxes were calculated by multiplying annual rBC means by annual snow accumulation. Annual snow accumulation was estimated based on our field measurements and the density profile from another 45-m-deep core drilled in the same area studied by Schwanck et al. (2016b) (described in section 4.1).

**253 **4. Results and discussion**

**254 **4.1** Core description**

During transport between Antarctica and the University of Maine the core was exposed to above-freezing temperatures and some sections were partially melted and refrozen. As the core was transported lying down in the boxes, this melt and refreeze occurred in the external part of the core and did not reach the center of it. The melted and refrozen portion of the core was removed by saw and hand scraping, and only a small 10-cm piece of section 07 was discarded as it was totally refrozen.

We used an ice core light table to observe the core stratigraphy. Millimeter-thick lenses of ice were observed all along the core, probably due to summer melting. Additionally, a few depth hoar layers up to 1 cm thick were observed. There were no visible dust layers.

The core density ranged from 0.38 to 0.60 g cm-3, not reaching the firn/ice transition of 264  $0.83 \text{ g cm}^{-3}$  (Fig. 2). We averaged the TT07 density profile with the density profile of another 265 core drilled in the same area of Antarctica (45 m deep; Schwanck et al., 2016b), fitted a 266 267 quadratic trend line to the average curve, and used this trend line to calculate the snow 268 accumulation, water equivalent (weq), and rBC fluxes for this work. We found an average snow accumulation of  $0.23 \pm 0.06$  weq m yr-1 for the entire core, so the 20.16-m length core 269 represents 10.65 weq m. For the 8 m analyzed in this work, the snow accumulation was  $0.21 \pm$ 270 271 0.04 weq m.

**272 **4.2** *Dating**

273 The first eight sections of the core, presented in this work, were dated to 17 years by annual-layer counting using mainly the rBC seasonal variability, as this is a reliable parameter 274 275 for dating (Winstrup et al., 2017). Data from Sentinel Hotspots indicate fires in Australia tend to peak in October, with the seasonal increase in fire activity occurring in August and the 276 277 decrease in December/January. The Programa Queimadas data show that fires in South 278 America tend to peak in September, with the seasonal increase in fire activity occurring in 279 June/July and the decrease in November/December. A comparison between the seasonality of 280 burning and the TT07 rBC record is presented in Fig. S3 in ESM.

As a support to this, we used sulfur (S), strontium (Sr) and sodium (Na), as these records show the more pronounced seasonal variability at the site (Schwanck et al., 2017), although we only had these analyzed down to ~7 m of the core. Also, the S, Sr and Na records are from a different core, retrieved a meter apart from the rBC core, and that core was subsampled and analyzed in another laboratory (CCI), meaning there could be some displacement from this record to the rBC one. The dating is presented in Fig. 3.

BC in Antarctica tends to peak during winter--spring (dry season) owing to drier conditions
in the Southern Hemisphere and a consequent increase in biomass burning (Bisiaux et al.,

289 2012b; Sand et al., 2017; Winstrup et al., 2017). Na and Sr also peak during this time, due to 290 intense atmospheric circulation and transport (Legrand and Mayewski, 1997), while S peaks in 291 late austral summer in relation to marine biogenic activity (Schwanck et al., 2017). We 292 considered our new year to match the end of what we define as the dry season, as this is a 293 reliable tying point in the record because of the abrupt drop in rBC concentrations based on the 294 fire-spot database from Australia and South America. This is also in agreement with Winstrup et al. (2017), who stated that rBC tends to peak a little earlier than New Year in their records 295 296 (Roosevelt Island Ice Core).

**297 **4.3** Nebulization efficiency**

298 The nebulization efficiency for the Marin-5 at CWU was calculated to be  $68.31\% \pm 5.91\%$ 299  $(1\sigma)$ , based on external calibration carried out every working day using the Aquadag standards 300 (see section 3.1). We found a decrease in nebulization efficiency during the laboratory work 301 period (-0.31%) per working day or -13.3% over the 43 working days), but we assume the 302 nebulization efficiency to have remained stable between the measurement of the standard and 303 the samples measured for the day, as in Katich et al. (2017). We attribute this decrease to the 304 Marin-5, but do not see any apparent cause. Pump flow rates were kept constant at  $0.14 \pm 0.02$ mL min-1 at all times during analysis. This result highlights the importance of making daily 305 306 Aquadag standards.

307 4.4 Whole-system repeatability

308 Samples were analyzed for 5 min each. Although a low particle count could increase the 309 uncertainty of the method, we noticed that the measurements did not vary significantly in 310 relation to analysis time, but much more so in relation to the sample average concentration 311 itself. To address this issue, we analyzed samples of varied rBC concentrations along the entire core more than once and for different periods of time. Each sample was analyzed between two and four times, for 5, 20 and/or 40 min. The samples were analyzed less than 2 h after melting to avoid rBC loss (Wendl et al., 2014).

While we observed no significant concentration variations for different analysis times (Fig. 4), our coefficient of variation (mean of all measurements of the sample × standard deviation) for concentrations lower than 0.03 µg L-1 was 25.7 ± 16.9 (1 $\sigma$ , *n* = 38), 10.4 ± 6.6 (1 $\sigma$ , *n* = 24) for concentrations between 0.03 and 0.07 µg L-1, and 7.3 ± 4.4 (1 $\sigma$ , *n* = 51) for concentrations higher than 0.07 µg L-1 (Fig. 5).

We attribute this variation to the number of collected particles in each sample: lowconcentration samples mean low particle triggers, which will lead to a higher variance in case rare particles large enough to contain a considerable fraction of total rBC mass are recorded.

324 **4.5** *rBC* concentrations and fluxes

We found a well-marked seasonal rBC cycle along the core (Fig. 6), with the same pattern of low summer/fall and high winter/spring concentrations as reported by Bisiaux et al. (2012b). As we collected our samples in January and the drilling was carried out from the snow surface, our core starts approximately in the New Year. As mentioned earlier, BC in Antarctica tends to peak during winter/spring, and so the New Year in the record is generally viewed as a steep decrease from peak concentrations to low concentrations. This was better observed in the CMS samples than the SSC ones for the 2014--15 transition.

Both sampling methods showed similar seasonality, but the CMS provided a smoother record (e.g., less summer/fall spikes) and a generally lower summer/fall concentration. Table 1 presents the details of this comparison. We attribute the smoother record to reduced handling of the core, as with SSC the individual samples were handled after decontamination to put them in the clean vials, which could have caused cross-contamination between samples to some degree. Nonetheless, a Wilcoxon--Mann--Whitney test indicated there to be no statistical difference between the two sample datasets at p = 0.01 (N = 650; two-tailed *P*-value = 0.449758; see Methods S1 in ESM).

For SSC, concentrations ranged from 0.003  $\mu$ g L-1 to 0.701  $\mu$ g L-1, with a geometric mean of 0.031  $\mu$ g L-1 (n = 307). Concentrations using the CMS ranged from < LOD (0.0015  $\mu$ g L-1) to 0.262  $\mu$ g L-1, with a geometric mean of 0.029  $\mu$ g L-1 (n = 343).

Summer/fall averages for both methods were also similar, with differences regarding 343 344 summer/fall highest values due to concentration peaks in the SSC method that did not alter the 345 mean significantly. Winter geometric means were similar for both methods (CMS =  $0.074 \mu g$  $L^{-1}$ ; SSC = 0.065 µg  $L^{-1}$ ); the winter maximum showed a pronounced difference owing to an 346 347 anomalous peak around the depth of 3 m, wherein the discrete sampling two consecutive samples achieved 0.701  $\mu$ g L-1 and 0.568  $\mu$ g L-1, while the continuous melter gave a maximum 348 of 0.147  $\mu$ g L-1 for the same depth. This almost five-fold difference did not appear anywhere 349 350 else in the core, probably reflecting contamination in the samples, and thus these two SSC 351 samples are not considered in further interpretations.

Figure 7 shows a dry- versus wet-season comparison for both methods. The results are similar for both methods: summer/fall values remain fairly steady for the entire record; winter/spring concentrations show an initial peak in 1998 and 1999 AD, followed by a low in 2002 and an increasing trend from 2002 to 2014---more visible in the CMS record (but with a weak  $r^2$  of 0.2478, not shown).

Annual rBC fluxes were calculated to account for potential biases in annual rBC concentrations due to changes in snow accumulation rates. Fluxes were calculated by multiplying annual rBC concentrations by the annual snow accumulation. rBC annual concentrations were averaged from SSC and the CMS. Concentrations and fluxes followed a 361 similar pattern, implying low variability in snow accumulation during the study period (Fig.362 8).

**363 **4.6** Comparison with other rBC cores in Antarctica**

Table 2 compares our results with other rBC records in Antarctica. East Antarctica cores [NUS0X from Bisiaux et al. (2012a)] present the highest elevations and annual rBC concentrations, but the lowest snow accumulation, in recent times (~1800--2000). The authors found a linear positive correlation between site elevation and rBC concentrations for the NUS07 cores, of 0.025  $\mu$ g L-1 (500 m)-1, and hypothesized that rBC inputs to the atmosphere over East Antarctica are not controlled by the intrusion of marine air masses and that transport in the upper troposphere may be more important.

Arienzo et al. (2017) found an even higher annual rBC concentration for the coastal site B40 (0.3  $\mu$ g L-1), where the flux was calculated to be 20  $\mu$ g m-2 yr-1. As BC is primarily deposited through wet deposition (Flanner et al., 2007), the authors attributed the higher accumulation in coastal areas to the scavenging of most of the BC, with fluxes lowering inland as the accumulation rates decreased.

Arienzo et al. (2017) also found high rBC fluxes for the WAIS ice core for the end of the last glaciation termination (14--12 k BP, 25  $\mu$ g m-2 yr-1) and for the mid-Holocene (12--6 k BP, 45  $\mu$ g m-2 yr-1). The authors attributed the high rBC fluxes in the past to a period of relatively high austral-burning-season and low growing-season insolation.

The WAIS ice core (Bisiaux et al., 2012b; Arienzo et al., 2017) is the closest to TT07 (350 km apart). Although the annual snow accumulation is similar at both sites ( $0.21 \pm 0.04$  weq m yr-1 for TT07 in this work;  $0.20 \pm 0.03$  weq m-1 for WAIS), our annual rBC concentration is less than half that of WAIS during 1850--2001 ( $0.031 \mu g L^{-1}$  for TT07;  $0.08 \mu g L^{-1}$  for WAIS). The rBC flux is also lower ( $6.1 \mu g m^{-2} yr^{-1}$  for TT07;  $16 \mu g m^{-2} yr^{-1}$  for WAIS), although we acknowledge there is not a large temporal overlap between the cores (three years, 1998--2001).

**5.** Conclusions**

387 This study shows that the CMS with discrete sampling is a faster and more reliable way of analyzing low-dust content samples compared with SSC, despite samples sitting in the liquid 388 389 state for a longer period of time (maximum of 1 h for SSC versus 2 h for the CMS). A long 390 sample waiting-time in the liquid state is normally not recommended because of the possible 391 changes in rBC concentrations caused by particle adhesion to the vial walls and the 392 agglomeration of particles outside the SP2 detection range (Wendl et al., 2014). However, in 393 this work, the longer time did not reflect any significant changes in rBC concentrations. The 394 CMS record was smoother than the SSC record, probably due to the reduced handling of the 395 snow and firn core during sub-sampling. SSC, though, needs much less volume than the CMS, 396 which could be an advantage when working with limited resources (samples). A Wilcoxon--397 Mann--Whitney test indicated there to be no statistical difference between the results of the 398 different methods at p = 0.01.

The record for these first 8 m of the snow and firn core shows a well-defined seasonal signal, with high rBC concentrations during the dry season (austral winter/spring) and low concentrations during the wet season (austral summer/fall). Both methods were able to identify these variations in rBC.

The TT07 core showed an annual rBC concentration below those of all other rBC cores in Antarctica referenced in this work, and fluxes similar to high-elevation East Antarctica ice cores (Bisiaux et al., 2012a).

Further studies addressing airmass trajectories are necessary to understand this. Arienzo et al. (2017) related the BC input to the WAIS core site to the intrusion of marine air masses, in which case coastal areas should have higher BC concentrations. Bisiaux et al. (2012a) suggested that transport in the upper troposphere may be more important in East Antarctica, in which case higher-elevation sites would show higher BC concentrations. As the TT07 site is

| 411 | located at higher elevation than the WAIS core (2122 MSL versus 1766 MSL, respectively),   |
|-----|--------------------------------------------------------------------------------------------|
| 412 | but has lower rBC concentrations and fluxes than WAIS, we postulate that the deposition of |
| 413 | BC at the site is more related to marine air masses than to upper-tropospheric transport.  |

*Acknowledgements.* This research is part of the Brazilian Antarctic Program
(PROANTAR) and was financed with funds from the Brazilian National Council for Scientific
and Technological Development (CNPq) Split Fellowship Program (Grant No. 200386/20182) and from the CNPq projects 465680/2014-3 and 442761/2018-0. We thank the Centro Polar
e Climático (CPC/UFRGS) and the Department of Geological Sciences (CWU) faculty and
staff for their support of this work. We also thank the anonymous reviewers for their comments
and suggestions, as well as the *Advances in Atmospheric Sciences* team.

| 421 | REFERENCES                                                                                   |
|-----|----------------------------------------------------------------------------------------------|
| 422 | Andela, N., and Coauthors, 2017: A human-driven decline in global burned area. Science, 356, |
| 423 | 13561362, https://doi.org/10.1126/science.aal4108.                                           |

424 Arienzo, M. M., J. R. McConnell, L. N. Murphy, N. Chellman, S. Das, S. Kipfstuhl, and R.

425 Mulvaney, 2017: Holocene black carbon in Antarctica paralleled Southern Hemisphere

426 climate. J. Geophys. Res., **122**, 6713--6728, https://doi.org/10.1002/2017JD026599.

Bice, K., and Coauthors, 2009: Black carbon: A review and policy recommendations.
Woodrow Wilson School of Policy & International Affairs. [Available online at http://www.wws.princeton.edu/research/PWReports/F08/wws591e.pdf]

430 Bisiaux, M. M., R. Edwards, J. R. McConnell, M. R. Albert, H. Anschütz, T. A. Neumann, E.

- 431 Isaksson, and J. E. Penner, 2012a: Variability of black carbon deposition to the East
- 432 Antarctic Plateau, 1800-2000 AD. *Atmospheric Chemistry and Physics*, **12**, 3799--3808,
- 433 https://doi.org/10.5194/acp-12-3799-2012.

- 434 Bisiaux, M. M., and Coauthors, 2012b: Changes in black carbon deposition to Antarctica from
- 435 two high-resolution ice core records, 1850--2000 AD. *Atmospheric Chemistry and Physics*,

436 **12**, 4107--4115, https://doi.org/10.5194/acp-12-4107-2012.

- 437 Bond, T. C., and Coauthors, 2013: Bounding the role of black carbon in the climate system: A
- 438 scientific assessment. J. Geophys. Res., 118, 5380--5552,
  439 https://doi.org/10.1002/jgrd.50171.
- Casey, K. A., S. D. Kaspari, S. M. Skiles, K. Kreutz, and M. J. Handley, 2017: The spectral
  and chemical measurement of pollutants on snow near South Pole, Antarctica. *J. Geophys. Res.*, 122, 6592--6610, https://doi.org/10.1002/2016JD026418.
- Flanner, M. G., C. S. Zender, J. T. Randerson, and P. J. Rasch, 2007: Present-day climate
  forcing and response from black carbon in snow. *J. Geophys. Res.*, 112, D11202,
  https://doi.org/10.1029/2006JD008003.
- Fretwell, P., and Coauthors, 2013: Bedmap2: Improved ice bed, surface and thickness datasets
  for Antarctica. *The Cryosphere*, 7, 375--393, https://doi.org/10.5194/tc-7-375-2013.
- 448 Hansen, J., and L. Nazarenko, 2004: Soot climate forcing via snow and ice albedos.
- 449 Proceedings of the National Academy of Sciences of the United States of America, 101,

450 423--428, https://doi.org/10.1073/pnas.2237157100.

- 451 Kaspari, S. D., M. Schwikowski, M. Gysel, M. G. Flanner, S. Kang, S. Hou, and P. A. 452 Mayewski, 2011: Recent increase in black carbon concentrations from a Mt. Everest ice 453 core spanning 1860--2000 AD. Geophys. Res. Lett.. 38, L04703, 454 https://doi.org/10.1029/2010GL046096.
- Kaspari, S., S. M. Skiles, I. Delaney, D. Dixon, and T. H. Painter, 2015: Accelerated glacier
  melt on Snow Dome, Mount Olympus, Washington, USA, due to deposition of black
  carbon and mineral dust from wildfire. *J. Geophys. Res.*, 120, 2793--2807,
  https://doi.org/10.1002/2014JD022676.

- Kaspari, S., T. H. Painter, M. Gysel, S. M. Skiles, and M. Schwikowski, 2014: Seasonal and
  elevational variations of black carbon and dust in snow and ice in the Solu-Khumbu, Nepal
  and estimated radiative forcings. *Atmospheric Chemistry and Physics*, 14, 8089--8103,
  https://doi.org/10.5194/acp-14-8089-2014.
- Katich, J. M., A. E. Perring, and J. P. Schwarz, 2017: Optimized detection of particulates from
  liquid samples in the aerosol phase: Focus on black carbon. *Aerosol Science and Technology*, 51, 543--553, https://doi.org/10.1080/02786826.2017.1280597.
- Koch, D., T. C. Bond, D. Streets, N. Unger, and G. R. van der Werf, 2007: Global impacts of
  aerosols from particular source regions and sectors. *J. Geophys. Res.*, 112, D02205,
  https://doi.org/10.1029/2005JD007024.
- Lauk, C., and K. H. Erb, 2009: Biomass consumed in anthropogenic vegetation fires: Global
  patterns and processes. *Ecological Economics*, **69**, 301--309,
  https://doi.org/10.1016/j.ecolecon.2009.07.003.
- 472 Legrand, M., and P. Mayewski, 1997: Glaciochemistry of polar ice cores: A review. *Rev.*473 *Geophys.*, 35, 219--243, https://doi.org/10.1029/96RG03527.
- Limpert, E., W. A. Stahel, and M. Abbt, 2001: Log-normal distributions across the sciences:
  Keys and clues: On the charms of statistics, and how mechanical models resembling
  gambling machines offer a link to a handy way to characterize log-normal distributions,
  which can provide deeper insight into variability and probability---normal or log-normal:
  That is the question. *BioScience*, **51**, 341--352, https://doi.org/10.1641/00063568(2001)051[0341:lndats]2.0.co;2.
- 480 Marlon, J. R., and Coauthors, 2016: Reconstructions of biomass burning from sediment-
- 481 charcoal records to improve data-model comparisons. *Biogeosciences*, **13**, 3225--3244,
- 482 https://doi.org/10.5194/bg-13-3225-2016.

- Marlon, R. J., and Coauthors, 2008: Climate and human influences on global biomass burning
  over the past two millennia. *Nature Geoscience*, 1, 697--702,
  https://doi.org/10.1038/ngeo313.
- 486 Matsuoka, K., A. Skoglund, and G. Roth, 2018: Quantarctica [Data set].
  487 https://doi.org/10.21334/npolar.2018.8516e961.
- McConnell, R. J., and Coauthors, 2007: 20th-century industrial black carbon emissions altered
  arctic climate forcing. *Science*, **317**, 1381--1384.
- 490 Mori, T., N. Moteki, S. Ohata, M. Koike, K. Goto-Azuma, Y. Miyazaki, and Y. Kondo, 2016:
- 491 Improved technique for measuring the size distribution of black carbon particles in liquid
- 492 water. Aerosol Science and Technology, 50, 242--254,
  493 https://doi.org/10.1080/02786826.2016.1147644.
- Moteki, N., and Y. Kondo, 2010: Dependence of laser-induced incandescence on physical
  properties of black carbon aerosols: Measurements and theoretical interpretation. *Aerosol Science and Technology*, 44, 663--675, https://doi.org/10.1080/02786826.2010.484450.
- 497 Olfert, J. S., J. P. R. Symonds, and N. Collings, 2007. The effective density and fractal
  498 dimension of particles emitted from a light-duty diesel vehicle with a diesel oxidation
  499 catalyst. *Journal of Aerosol Science*, 38, 69--82,
  500 https://doi.org/10.1016/j.jaerosci.2006.10.002.
- 501 Osmont, D., M. Sigl, A. Eichler, T. M. Jenk, and M. Schwikowski, 2018a: A Holocene black
  502 carbon ice-core record of biomass burning in the Amazon Basin from Illimani, Bolivia.
- 503 *Climate of the Past*, **15**, 579--592, https://doi.org/10.5194/cp-15-579-2019.
- 504 Osmont, D., I. A. Wendl, L. Schmidely, M. Sigl, C. P. Vega, E. Isaksson, and M. Schwikowski,
- 505 2018b: An 800-year high-resolution black carbon ice core record from Lomonosovfonna,
- 506 Svalbard. Atmospheric Chemistry and Physics Discussions, https://doi.org/10.5194/acp-
- 507 2018-244. (in press)

- 508 Osterberg, C. E., M. J. Handley, S. B. Sneed, P. A. Mayewski, and K. J. Kreutz, 2006: 509 Continuous ice core melter system with discrete sampling for major ion, trace element, 510 Environ. Sci. Technol., 40, 3355--3361, and stable isotope analyses. 511 https://doi.org/10.1021/es052536w.
- 512 Petzold, A., and Coauthors, 2013: Recommendations for reporting black carbon measurements.
- 513 Atmospheric Chemistry and Physics, 13, 8365--8379, https://doi.org/10.5194/acp-13514 8365-2013.
- Sand, M., and Coauthors, 2017: Aerosols at the poles: An AeroCom Phase II multi-model
  evaluation. *Atmospheric Chemistry and Physics*, **17**, 12197--12218,
  https://doi.org/10.5194/acp-17-12197-2017.
- Schwanck, F., J. C. Simões, M. Handley, P. A. Mayewski, R. T. Bernardo, and F. E. Aquino,
  2016a: Anomalously high Arsenic concentration in a West Antarctic ice core and its
  relationship to copper mining in Chile. *Atmos. Environ.*, **125**, 257--264,
  https://doi.org/10.1016/j.atmosenv.2015.11.027.
- Schwanck, F., J. C. Simões, M. Handley, P. A. Mayewski, R. T. Bernardo, and F. E. Aquino,
  2016b: Drilling, processing and first results for Mount Johns ice core in West Antarctica
  Ice Sheet. *Brazilian Journal of Geology*, 46, 29--40, https://doi.org/10.1590/2317-
- 5254889201620150035.
- 526 Schwanck, F., J. C. Simões, M. Handley, P. A. Mayewski, J. D. Auger, R. T. Bernardo, and F.
- 527 E. Aquino, 2017: A 125-year record of climate and chemistry variability at the Pine Island
- 528 Glacier ice divide, Antarctica. *The Cryosphere*, **11**, 1537--1552,
  529 https://doi.org/10.5194/tc-11-1537-2017.
- Sigl, M., N. J. Abram, J. Gabrieli, T. M. Jenk, D. Osmont, and M. Schwikowski, 2018: 19th
  century glacier retreat in the Alps preceded the emergence of industrial black carbon

532 deposition on high-alpine glaciers. *The Cryosphere*, **12**, 3311--3331,
533 https://doi.org/10.5194/tc-12-3311-2018.

- Stohl, A., and H. Sodemann, 2010: Characteristics of atmospheric transport into the Antarctic
  troposphere. J. Geophys. Res., 115, D02305, https://doi.org/10.1029/2009JD012536.
- Tao, G. H., R. Yamada, Y. Fujikawa, A. Kudo, J. Zheng, D. A. Fisher, and R. M. Koerner,
  2001: Determination of trace amounts of heavy metals in arctic ice core samples using
  inductively coupled plasma mass spectrometry. *Talanta*, 55, 765--772,
  https://doi.org/10.1016/S0039-9140(01)00509-4.
- Wang, Z., J. Chappellaz, K. Park, and J. E. Mak, 2010: Large variations in southern hemisphere
  biomass burning during the last 650 years. *Science*, 330, 1663--1666,
  https://doi.org/10.1126/science.1197257.
- Wendl, I. A., J. A. Menking, R. Färber, M. Gysel, S. D. Kaspari, M. J. G. Laborde, and M.
  Schwikowski, 2014: Optimized method for black carbon analysis in ice and snow using
- 545 the Single Particle Soot Photometer. *Atmospheric Measurement Techniques Discussions*,

546 **7**, 3075--3111, https://doi.org/10.5194/amtd-7-3075-2014.

- 547 Winstrup, M., and Coauthors, 2017: A 2700-year annual timescale and accumulation history
- 548 for an ice core from Roosevelt Island, West Antarctica. *Climate of the Past Discussions*,
- 549 https://doi.org/10.5194/cp-2017-101. (in press)

551 Table 1. Main results from the comparison between SSC and the CMS. All values are in units of  $\mu g L^{-1}$ . "Geomean" refers to the geometric mean, and  $1\sigma^*$  is the multiplicative standard 552 deviation, representing 68.3% of the variability (Limpert et al., 2001; Bisiaux et al., 2012a). 553

554 555 556

|                      | (a) Total a |                           |
|----------------------|------------------------|---------------------------|
|                      | Solid-state sampling   | Continuous melting system |
| Geomean              | 0.031                  | 0.029                     |
| $1\sigma^*$ interval | 0.0130.073             | 0.0110.076                |
| Lowest/highest conc. | 0.003/0.701            | 0.001/0.262               |
|                      | (b) Wet-season         |                           |
|                      | Solid-state sampling   | Continuous melting system |
| Geomean              | 0.019                  | 0.016                     |
| $1\sigma^*$ interval | 0.0110.032             | 0.0080.027                |
| Lowest/highest       | 0.003/0.083            | 0.001/0.071               |
|                      | (c) Dry-season         |                           |
|                      | Solid-state sampling   | Continuous melting system |
| Geomean              | 0.065                  | 0.074                     |
| $1\sigma^*$ interval | 0.0290.121             | 0.0350.128                |
| Lowest/highest       | 0.007/0.701            | 0.014/0.262               |
|                      |                        |                           |

aAll samples from section 1 of the core (surface) down to section 8 (around 8 m deep). Total number for solidstate sampling is 307, and for continuous melting system is 343.

560 Table 2. Coordinates, elevation, period covered and BC information for this study and previous 561 works on Antarctic ice cores. To enable direct comparison, we only list studies that used the SP2. 562

| Source                       | Core
name | Locati
on in
Antarc
tica | Lat./
Long.               | Elev.
(MS
L) | Period
covered | Annual
BC
conc.
(µg L -1 ) | BC
conc.
range
$(2\sigma)^{a}$ | Annual
accum.
(weq m
yr -1 ) | Annual
BC
fluxes
$(\mu g m^{-2} y r^{-1})$ | BC flux range $(2\sigma)$ |
|------------------------------|--------------|-----------------------------------|------------------------------|--------------------|-------------------|------------------------------------------------|-----------------------------------------|--------------------------------------------------|-----------------------------------------------------|---------------------------|
| This
study                | TT07         | West                              | 79°55'S
,
94°21'
W  | 2122               | 1998
2015      | 0.029                                          | 0.01
0.07                            | 0.21 ± 0.04                                      | 6.1                                                 | 2.6
14.6               |
| Bisiaux
et al.
(2012b) | WAIS         | West                              | 79°46'S
,
112°08'
W | 1766               | 1850
2001      | 0.08                                           | 0.05
0.12                            | 0.20 ± 0.03                                      | 16                                                  | 9.8
24.4               |
|                              | Law
Dome  | East                              | 66°73'S
,
112°83'
E | 1390               | 1850
2001      | 0.09                                           | 0.05
0.2                             | 0.15 ± 0.03                                      | 13.5                                                | 7.3
30.6               |
| Arienzo
et al.            | WAIS         | West                              | 79°46'S
,                 | 1766               | 1412 k
BP      | 0.12 b                              | -                                       | -                                                | 25 b                                     | -                         |
| (2017)                       |              |                                   | 112°08'
W                 |                    | 126k
BP        | 0.2 b                               | -                                       | -                                                | 45 b                                     | -                         |
|                              | B40          | East                              | 70°0'S,
0°3'E             | 2911               | 2.5k0
BP       | 0.3°                                           | -                                       | -                                                | 20°                                                 | -                         |
| Bisiaux
et al.
(2012b) | NUS0
7-1  | East                              | 73°43'S
,
07°59'E      | 3174               | 1800
2006      | 0.16                                           | 0.09
to
0.26                      | 0.05±0.
02                                    | 8.3                                                 | 4.6
to
14.2         |
|                              | NUS0
7-2  |                                   | 76°04'S
,
22°28'E      | 3582               | 1800
1993      | 0.12                                           | 0.07
0.19                            | $\begin{array}{c} 0.03 \pm \\ 0.01 \end{array}$  | 3.9                                                 | 2.5
6.2                |
|                              | NUS0
7-5  |                                   | 78°39'S
,
35°38'E      | 3619               | 1800
1989      | 0.14                                           | 0.08
0.26                            | $\begin{array}{c} 0.02 \pm \\ 0.01 \end{array}$  | 3.4                                                 | 1.8
6.3                |
|                              | NUS0
7-7  |                                   | 82°49'S
,
54°53'E      | 3725               | 1800
2008      | 0.18                                           | 0.12
0.27                            | $\begin{array}{c} 0.02 \pm \\ 0.01 \end{array}$  | 5.3                                                 | 3.5
8.0                |
|                              | NUS0
8-4  |                                   | 82°49'S
,
18°54'E      | 2552               | 1800
2004      | 0.1                                            | 0.06
0.18                            | $\begin{array}{c} 0.04 \pm \\ 0.01 \end{array}$  | 3.7                                                 | 2.1
6.9                |
|                              | NUS0
8-5  |                                   | 82°38'S                      | 2544               | 1800
1993      | 0.11                                           | 0.07
0.18                            | $\begin{array}{c} 0.03 \pm \\ 0.01 \end{array}$  | 3.9                                                 | 2.2
6.5                |

a Multiplicative standard deviation representing 95.5% of the confidence interval b 50-year average, not annual c 7-years media, not annual

568 **Fig. 1.** Drilling location for the snow and firn core analyzed in this work (TT07) and other BC

569 cores mentioned in the text. The bottom-left inset shows the drilling site in perspective to South

570 America. Base map from the Quantarctica Project (Matsuoka et al., 2018).

Fig. 2. Density profile of the snow and firn core analyzed. Depth is presented in meters and
water equivalent (weq) meters. The quadratic fit was calculated from the average density
profile from this work and from Schwanck et al. (2016b).

---

## Author Response (AR1)

**Author's response to the editor and reviewers**

Dear TC editor and referees,

Please find below the point-by-point response to the reviews including a list of all relevant changes and a marked-up manuscript version. Please note the responses to the referees presented here are the same responses as submitted on February 27th - we added them in this .pdf file for your convenience.

Response to Anonymous Referee #1 comment
Response to Anonymous Referee #2 comment
Marked-up Manuscript

We hope all raised questions have been responded, and are looking forward the re-review.

We thank you for your time reviewing our manuscript.

Luciano Marquetto (first author)

**Response to Anonymous Referee #1 comment**

*Dear Anonymous Referee,*
*We thank you for your time, expertise, and helpful suggestions.*

*We apologize for the inconvenience that Marquetto et al. (2019) was not available during the review process. Marquetto et al. (2019) was accepted for publication in October 2019 and will be available online in April 2020 in "Advances in Atmospheric Sciences" as Marquetto et al. (2020). We have added the accepted manuscript as supplementary material for your review. Considering our system setup and calibrations carried out for the SP2, we believe the rBC concentrations found in this work are solid results, as described with more details along the text.*

*We made several improvements to the manuscript based on the reviewer's suggestions. First, we opted to remove the 2008 trace element records from the dating section. Although the 2015 and 2008 cores presented an overlap of 7 years (2002-2008), there is a 20% distortion between them in order to match both. This raised an issue about using a core 850 m away, with topographical differences, to date the rBC core. We decided this brings more uncertainty to our work than using the 2015 trace element record to constrain the dating down to 2002, and then base the rest of the dating on the relations of rBC, Na, Sr and S observed for the 2015-2002 period.*

*We also removed the sodium record from the spectral analysis and comparison of rBC and Na transport, as the 2015-2002 Na record is too short to show cycles in the spectrum.*

*On the other hand, we added atmospheric transport simulations using the HYSPLIT model to identify BC source areas. Results corroborated our initial conclusions of Australia and New Zealand as the most probable sources of BC to the TT07 drilling site, and indicated limited influence of South American air masses. More information is presented at the end of this document, after responses to reviewer's suggestions.*

*We also added a comparison of the Antarctic rBC records with snow accumulation, elevation and distance from open sea. In East Antarctica, rBC concentrations have a negative correlation with snow accumulation and positive correlation with elevation and distance to the sea, whereas in West Antarctica rBC concentrations present a positive correlation with snow accumulation and a negative correlation with elevation and distance to the sea. These opposite trends may indicate differences in rBC transport to East and West Antarctica. While for East Antarctica upper tropospheric transport and dry deposition may be the main controllers of rBC concentrations (Bisiaux et al., 2012b), for*

*West Antarctica rBC concentrations may be modulated by intrusion of air masses from the marine boundary layer, contrary to what was previously suggested (Bisiaux et al., 2012a). Low elevations in West Antarctica facilitates the intrusion of moisture-rich cyclones and the transport of aerosols inland (Neff and Bertler, 2015; Nicolas and Bromwich, 2011), while the positive relationship between West Antarctica rBC concentrations and snow accumulation may indicate rBC to be primarily deposited through wet deposition, being scavenged along the coastal regions were snow accumulation is higher. More information is presented at the end of this document, after responses to reviewer's suggestions.*

*The rBC record we present in this work is from an unique area – the Pine Island/Institute Glacier Divide, where air masses from the Weddel and Bellingshausen Seas converge (Parish and Bromwich, 2007), and is the highest altitude rBC core collected in West Antarctica. Considering all these improvements and new findings, we believe our manuscript is suitable for The Cryosphere.*

*Please find our responses in italic, while we kept your original comments in normal text.*

**Original referee comment:**

This manuscript by Marquetto et al presents a 47 yr black carbon record from a 20- m firn core recovered from the Pine Island Glacier in West Antarctica. The authors measured rBC using the SP2 method, and dated the core primarily using the seasonal cycle of rBC. Potential southern hemisphere rBC source regions to this site were explored by correlating the seasonal cycle of rBC to Australian and South American fire spot data and GFED biomass burning emissions estimates as well as by comparing the power spectrum of rBC to large-scale atmospheric patterns (ENSO, AAO, and ASL) and fire spot data.

While Antarctic rBC records are important for understanding changes in southern hemisphere biomass burning as well as radiative forcing, and undoubtedly an immense amount of work went into developing this highly-resolved dataset, the interpretation and discussion are not thorough or novel enough to add significantly to the understanding of rBC deposition in West Antarctica. The discussion focused on three analyses: SNICAR snow albedo modeling, identifying continental emissions sources, and linking rBC to atmospheric circulation using spectral analysis. The SNICAR modeling showed that rBC deposition at this site has little to no effect on snow albedo, which is not unexpected given the extremely low rBC concentrations and (as noted by the authors) has already been shown for clean Antarctic snow (Casey et al., 2017). The identification of source regions by comparing seasonal cycles in observed rBC and GFED fire emissions is purely correlativeă˘Ă˘ TI do not think is a strong enough approach, especially with my concerns about dating, to draw conclusions about source regions without a more robust approach that would consider atmospheric transport and magnitude of biomass

burning emissions. Finally, the spectral analysis was similar to that conducted by Bisiaux et al. (2012) for the WAIS Divide and Law Dome rBC records and does not provide any concrete new links between atmospheric circulation and Antarctic rBC. Furthermore, I have concerns about the factor of 2-3 lower rBC concentrations compared to other West Antarctic ice cores (see comments below).

Overall, since the conclusions do not add substantial new insight or understanding to rBC deposition and mainly confirm what is already known, I do not think this manuscript is suitable for the scope of The Cryosphere.

I have two major concerns about the methods: 1.) rBC measurements, and 2.) dating.

*We respond to the two major concerns below.*

1. rBC measurements: The rBC concentrations presented in this study are a factor of 2-3 lower than rBC concentrations in other West Antarctic cores, including the WAIS Divide ice core as well as other measurements from Pine Island Glacier (Pasteris et al., 2014). Note that rBC measurements from early 1900s to 2006 from Pine Island Glacier (as well as Thwaites Glacier and the divide between Pine Island and Thwaites Glaciers) were previously published in Pasteris et al. (2014) and are also 2-3x higher than the concentrations presented in this manuscript- it would be worth including this citation and even comparing to this published dataset to see how the magnitude and temporal variability of the records compare.

*We included Pasteris et al. (2014) as a citation along our manuscript and added their data as comparison in section 4.4. Thank you for pointing this out.*

The authors need to provide more information to determine if this offset is real or a result of the SP2 calibration/analytical system. Since the referenced Marquetto et al. (2019) manuscript does not appear to be published at the time of this review, please include details on the SP2 internal/external calibration in this manuscript.

*We believe the lower rBC concentrations found by this work are not due to less-than-optimal SP2/analytical setting, and are, in fact, a reflection of the true rBC concentration in the samples. We were diligent in maintaining low background concentrations for this study, and detailed specifics on the methodology are provided in this response and in the revised manuscript.*

*We attached Marquetto et al. (2020) so you can check the internal and external calibration thoroughly. In short, internal calibration was carried out using a Centrifugal Particle Mass Analyzer (CPMA) to select 23 particle masses from 0.5 fg to 800 fg from an Aquadag solution. Each selected mass ran for 30 min to 6 h to provide statistically significant particle triggers to calibrate the SP2, and calibration curves were then generated for all SP2 channels. The data presented in the manuscript in review is from the duplicated extended range broadband detector (B2HG), as this channel gave the*

*best-fit calibration curve (spline) of all channels in the 0.5 – 200 fg range, but with an upper detection limit at ~ $D_{BC}$ = 600 nm. The external calibration was carried out daily using five fresh Aquadag standards ranging from 0.01 to 1.0 µg $L^{-1}$ and one environmental standard (diluted meltwater of a snow sample from Table Mountain, WY, USA) of known concentration (0.18 ± 0.04 µg $L^{-1}$).*

[Figure]

**Calibration curves for the (a) B2HG channel (0.5 < M < 10 fg – linear scale), (b) B2HG channel (0.5 < M < 200 fg – logarithmic scale) obtained from the internal calibration carried out using the SP2+CPMA.**

*Please note that Marquetto et al. (2020) present a comparison between two subsampling methods (solid state cutting and continuous melter system) – both resulted in statistically similar rBC concentrations.*

How were the ice samples melted (room temperature?), and how soon before SP2 analysis were they melted (lines 96-97)? Wendl et al. (2014) show how rBC is lost after melting during sample storage in polypropylene vials over just a few days (with proportionally greater losses for low-concentration samples).

*We added in section 3.3 the sentence: "Samples were melted at room temperature or in a tepid bath not exceeding 25°C, sonicated for 15 min, and then analyzed (in less than 1 h after melting)." This is also described in (Marquetto et al., 2020).*

How often was 5% HNO3 used to clean the system (line 118), and how did you determine when the acid was flushed from the system? Wendl et al. (2014) also discuss how acidification can result in significant loss of rBC.

*The results regarding acidification presented in Wendl et al. 2014 were conducted at Central Washington University, where we conducted this research, so we are well aware of these findings. Rather, we were very diligent to maintain the system at the MQ background level (at 0-0.5 particles cm$^{-3}$, translating to less than 0.01 µg L$^{-1}$ rBC concentration). When MQ levels increased to 0.5 particles cm$^{-3}$, the HNO3 was used to clean the TruFlo liquid flow monitor and the nebulized. The HNO3 was only used on three days during the 43 working days, sometimes more than once a day. We ran the solution for 10 to 30 min from the peristaltic pump to the nebulizer outlet tubing (disconnected from the SP2). After using the acid we ran MQ water (MilliQ-Element, Millipore, 18.2 MΩ cm) for at least double the time we ran the acid. After connecting the SP2 back, we ran MQ water for at least 15 min. These are the procedures used in the Ice Core Laboratory (CWU), where the samples were analyzed.*

What kind of tubing was used to transport the aerosol from the Marin 5 nebulizer to the SP2?

*We used black conductive tubing from Simolex Rubber Corporation (Plymouth, USA) - http://simolex.com/product/conductive-silicone-extrusion-products-tubing/. This is the standard tubing that comes with the SP2.*

2. The description of the dating, namely the role of the datasets from the Schwanck et al. (2017) study, is vague. Since much of the analysis, including the seasonal cycle correlations and spectral analysis, require precise dating, more explanation of the dating must be given. The annual picks in Fig. 1 are consistently on the austral summer (January) decrease of rBC concentration, but appear to be inconsistently placed across the S, Sr, and Na records.

*We opted to use only the 2015 trace element record for dating and constrain the rest of the dating based on the rBC well defined seasonality for West Antarctica (Arienzo et al., 2017; Bisiaux et al., 2012a; Winstrup et al., 2017) and for the Pine Island Glacier (Pasteris et al., 2014).*

*To improve our dating for these first meters we reviewed sample resolution for the rBC and trace element cores, and added an additional parameter to dating: the maxima in the non-sea-salt sulfur to sodium (nssS/Na) ratio, a robust seasonal indicator that peaks around the new year (Arienzo et al., 2017). This parameter helps in the identification of the annual layers more than the Na and S records alone. Non-sea-salt sulfur was calculated using Eq. 3 to 6 from Schwanck et al. (2017) and references therein.*

*We consider this dating to have ±2 years uncertainty. The first uncertain year is located at 6.18 m (between 2003 and 2002, figure 2a), where S and nssS/Na peak but no full cycle is observed in the rBC record. We did not consider this to be a year, as rBC does not present a full cycle. The second uncertain year is located at 18.14 m (year 1973, figure 2b) where there is no clear rBC peak but snow accumulation would be*

*anomalously high if considered to be only a year instead of two. We consider this to be an annual pick and consequently two years, as there is no evidence of higher-than-normal snow accumulation in the region for this period* (Kaspari et al., 2004).

[Figure]

**Figure 2. (a) Dating of the snow and firn core based on rBC and using S, Sr, Na and nssS/Na records from nearby core (see section 3.6) as support for the first 6.5 meters. Dashed lines indicate estimated New Year and red dotted line indicate uncertainty in dating, explained in the text. (b) Dating for the full core (y axis logarithmic). Red dotted line indicates uncertainty in dating, as explained in the text.**

Is the S, Sr, and Na data shown in Fig. 2 all from the Schwanck et al. core from _1 km away?

*We opted to not use the 2008 record, which means the S, Sr and Na records in the next revision will be from the 2015 core collected a meter apart from the rBC record, down only to 6.5 m.*

I would expect cores 1 km apart to have a depth offset, so I would not be confident in correlating the picks from TT07 to the Schwanck et al. core without a common dataset to both cores to linking the cores in depth. Without showing that the chemical species from the TT07 core and Schwanck core are not offset in depth, it is not advisable to guide the dating of the TT07 core with data from the Schwanck et al. core, and likewise not justified to apply annual picks based on rBC in TT07 core to the Schwanck core Na record (I can't tell if the Na spectral analysis was conducted on the TT07 age scale or the original Schwanck et al. age scale). How does the Schwanck et al. data compare to the 5.6 m of S, Sr, and Na data for the core taken immediately adjacent to the rBC core (mentioned on lines 154-155)? There should be a few years of overlapping data to compare and that would at least be a start to justify comparing the two cores in depth.

*There is an overlap in the two cores, from 2008-2002, although a perfect match in peaks means a 20% distortion in the records (~0.2 water equivalent meters difference between the 2008 and 2015 cores). Due to this we decided not to use data from Schwanck et al. (2017) for dating anymore.*

Furthermore, there appears to be circular logic between the dating and seasonal comparison to GFED. If indeed the drop in rBC concentration at the end of the austral summer was used as the primary annual pick, and this pick was justified by GFED/fire spot seasonality as stated in lines 161-163, how can you then draw meaningful conclusions about the timing of rBC vs. GFED (in Fig. 6a) to identify source regions? The rBC and GFED are already inherently linked based on how you defined the dating of the core. Based on GFED seasonality in Fig. 6a, BC emissions drop in November for Asia/Africa/S. America, two months before the January drop for Australia/NZ. Even a month or two difference on where the rBC drop is assigned could have significant implications for the correlations used in section 4.6 (based on n=12 months) which are used to underpin the conclusion on lines 286-289. It would be much more appropriate to date the TT07 core using an independent chemical species, and then use the independent dating to examine the rBC seasonality.

*We do not see the dating as circular logic due to the use of GFED4s/fire spots data. We changed the text in section 3.6 to clarify this point:*

*"We considered the new year to match the end of what we define as the austral dry season, as this is a reliable tie point in the record due to the abrupt drop in rBC concentrations. Previous studies have demonstrated that rBC deposition occurs in*

*winter/spring, mostly September to December. For example: Arienzo et al. (2017) observed rBC concentrations to peak in September in the WAIS Divide ice core; Winstrup et al., (2017) used annual variations in rBC as the most reliable annual tracer for the Roosevelt Island Climate Evolution (RICE) ice core, stating that rBC tends to peak earlier in the year than January 1st. Pasteris et al. (2014) also corroborates rBC to peak in October and drop after for the Pine Island and Thwaites Glaciers, with lowest values from February to June. Bisiaux et al. (2012a) state that sub-annual rBC concentrations are highly seasonal in the WAIS Divide ice core for the period spanning 1850-2000 - low austral wet season and high austral dry season concentrations - and presented annual picks in the drop in rBC concentrations, as in this work. This is also consistent with the BC emission estimates from GFED4s and the fire spot databases from Australia and South America."*

*As for dating the core using independent chemical species, we used the trace element data as a secondary parameter to dating due to the offset that could exist between it and the rBC record. Samples are not co registered, they were not analyzed from the same core and same vial, so their resolution will be different. Using the independent chemical species to date the rBC record in monthly resolution would lead to greater uncertainty than using the rBC dating, especially considering rBC seasonality is already well defined in West Antarctica (Arienzo et al. 2017; Bisiaux et al., 2012; Winstrup et al., 2017; Pasteris et al. 2014).*

**Other comments**

Lines 14-15: Please specify what you mean by wet and dry season? I assume southern hemisphere wet/dry season, not at the ice core site (also on lines 145-146).

*We mean austral dry and wet season. Added the word "austral" in all dry/wet season citations.*

Lines 87 and 82: Is Marquetto et al. (2019) published and available?

*Marquetto et al. (2019) was accepted for publication in October 2019 and will be available online in April 2020 in "Advances in Atmospheric Sciences" as Marquetto et al. (2020). We have added the accepted manuscript as supplementary material for your review.*

Lines 83 and 90: Same heading title for sections 3.2 and 3.3.

*Corrected. Section 3.3 should be entitled: "Laboratory and vial cleaning".*

Line 116: Did you average the rBC data for the full 5 minutes that it was run? Can you quantify the stability of the measurement with a standard deviation over the time period averaged?

*The total rBC mass during the period of measurement, along with liquid and air flows, is used to calculate the rBC concentration for each sample using the Paul Scherrer Institute SP2 toolkit 4.200f. During sample analysis we monitored the samples to ensure that the incandescent particle concentration was stable. Liquid and air flows are also stable during analysis, as described in lines 108-115, leading to a robust measurement. We don't see the utility in in applying a standard deviation to the time period averaged, as the liquid concentration is based on the entire sampled period. The total mass of the measured particles is what is used in the calculation. For Antarctic samples, particularly the low concentration wet season (austral summer/fall), the concentrations are very low (0.015 ug/L). At these low concentrations particles are crossing the SP2 laser at a regular, but intermittent rate (i.e., the SP2 incandescent concentration can be stable, maintaining incandescent particle concentrations at 0-2 particles/cc during the analysis period).*

*We observed, though, that stability of the measurements depends more on sample rBC concentration than analysis time. Marquetto et al. (2020) presents this reproducibility test. We analyzed samples more than once and for different times (5, 20 and 40 minutes), and our coefficient of variation (mean of all measurements of the sample × standard deviation) did not vary with different measurement times, but was much higher for concentrations lower than 0.03 µg L$^{-1}$ (25.7 ± 16.9%) than for higher concentrations (for concentrations between 0.03 and 0.07 µg L$^{-1}$, it was 10.4 ± 6.6%, and for concentrations higher than 0.07 µg L$^{-1}$ it was 7.3 ± 4.4%.*

Lines 135-143: The Brazilian hotspot data is defined here, but never mentioned in the results and discussion section. Can it be omitted? Or did it result in a null finding?

*It resulted in a null finding. Line 318 briefly talks about that: "All other spectra showed only well-marked annual periodicities and intra annual periodicities of 2 and 3 cycles per year (0.5 and 0.3-year bands, not shown)."*

*We changed the sentence to: "All other spectra (including Programa Queimadas satellite data) showed only well-marked annual periodicities and intra annual periodicities of 2 and 3 cycles per year (0.5 and 0.3-year bands, not shown)."*

Line 140: How do the timeseries of the fire hotspot data compare to the rBC data? You only compare the power spectrums in this study. Do years with more hotspots correspond to years with more rBC deposition?

*This comparison can be seen in the figure below (Fig. S3 in Marquetto et al. (2020)). Years with more hotspots do not necessarily correspond to years with more rBC deposition. This is the reason why we compared the power spectrums, to identify increases and decreases in common for the datasets without looking only at the absolute values. rBC concentrations in Antarctica are not only a result of BC emissions, but also of atmospheric transport, deposition during transport and physical processes in the drilling site (Bisiaux et al. 2012), so we do not expect hotspots intensity to necessarily match years with more rBC deposition.*

[Figure]

Fig. S3 from Marquetto et al. (2020). TT07 rBC record (rescaled to monthly resolution) compared to Southern Hemisphere South America (SHSA) and Australian/New Zealand firespot records.

Lines 147-148: Please define Na, Sr, and S before using abbreviations

*Added definition in the abstract and in the manuscript (lines 14-15).*

Lines 167-173: Please plot the average of the two density profiles in Fig. 3 that the quadratic equation was fit to.

*Plotted. Density was averaged using curve fitting (polynomial terms = 3). Figure and caption are as follows:*

[Figure]

**Figure 3. TT07 density profile (blue). Depth is presented in meters and water equivalent (weq) meters. The quadratic fit was calculated from the average density profile (black) from this work and from Schwanck et al. (2016).**

Line 180: Please explicitly state which months went into the summer/fall and winter/spring averages.

*Added to the text:    summer/fall (wet season – January to June); winter/spring (dry season – July to December). We also corrected this specific line, where wet and dry season were attributed to the wrong annual seasons:*

*"We present our data as summer/fall ( wet season – January to June) concentrations and winter/spring ( dry season – July to December) concentrations."*

Line 193 and 296: Is the Na record you use for spectral analysis from the TT07 core, or is it from Schwanck et al.? If it is from Schwanck et al., have you changed the dating?

*We removed the sodium record from the spectral analysis and comparison of rBC and Na transport, as the 2015-2002 Na record is too short to show cycles in the spectrum.*

Lines 211-212: Per comments above, please clarify what dating is used for which chemical species.

*We removed the sodium record from the spectral analysis and comparison of rBC and Na transport, as the 2015-2002 Na record is too short to show cycles in the spectrum.*

Lines 244-246: Please include comparison to rBC data from Pasteris et al. (2014) from Pine Island Glacier.

*We included Pasteris et al. (2014) as a citation along our manuscript and added their data as comparison in section 4.4. Thank you for pointing this out.*

Line 338: Abstract says record extends from 1968-2015.

*Corrected the date to 1968-2015.*

Figure 1: Please check location of B40 ice core site.

*Corrected B40 site and added locations from Pasteris et al. (2014) – THW, DIV and PIG (see below)*

[Figure]

Figure 2: Please define which records are from the TT07 core and which are from Schwank et al.

As commented above, we decided not to present the 2008 core from Schwanck et al. (2017) for the dating.

Figures 4, 5: It would be helpful to include the standard deviation (or standard error) of the measurements that went into the annual averages and wet/dry season plots as an error bar.

*We added the geometric standard deviation (GSD) of annual and wet/dry rBC concentrations, as in the image and caption below. We did not include standard deviations in Figure 5 as the GSD for rBC annual average is already presented in Figure 4. We opted to present GSD in shaded area instead of error bars for readability, especially in the top graph.*

[Figure]

**Figure 4. (top) rBC concentrations for the entire core. Black thick line represents annual averages, while black thin line represents monthly values. (bottom) Dry season and wet season average concentrations per year. Note the y axis scale is logarithmic. Shaded areas in both top and bottom represent one geometric standard deviation of the monthly values.**

Figures 6, 8: Again, it would be helpful to have an estimate of uncertainty on the seasonal cycles.

*Added the geometric standard deviation to Figure 6 (for rBC), as below. We decided not to present the Na and rBC seasonality comparison anymore, as there is no point relating Na and rBC now that we are not talking about cycle differences in the spectral analysis.*

*Note that the geometric standard deviation is broad for rBC, which was expected considering concentrations are very low and seasonal peaks vary with time (e.g. the rBC dry season peak in 1999 is much higher than the 2002 dry season peak).*

[Figure]

**Figure 6(a). TT07 rBC (monthly averages, 1968-2014) and BC emissions estimated from GFED4s for the four SH regions (normalized, 1997-2014). Shaded area indicates rBC geometric standard deviation.**

Figure 7: Did you use rBC flux or concentration for the spectral analysis?

*We used rBC concentration. Added this information to the figure caption.*

Figure 8: Could the broad Na peak over 4 months be a result of mixing/matching Na data and depth picks between the TT07 and Schwanck et al. cores?

*As mentioned above, we are not comparing Na and rBC cycles anymore.*

Table 4: Please include Pasteris et al. (2014) rBC data.

*We included Pasteris et al. (2014) as a citation along our manuscript and added their data as comparison in section 4.4. Thank you for pointing this out.*

Additional information regarding HYSPLIT model:

We added atmospheric transport simulations from using the HYSPLIT model to identify BC source areas. We added two additional sections, one in methodology and another in results. They are as follows:
3. Methodology

[revised manuscript text omitted]

**Response to Anonymous Referee #2 comment**

*Dear Anonymous Referee,*

*We thank you for your time, expertise, and helpful suggestions.*

*We apologize for the inconvenience that Marquetto et al. (2019) was not available during the review process. Marquetto et al. (2019) was accepted for publication in October 2019 and will be available online in April 2020 in "Advances in Atmospheric Sciences" as Marquetto et al. (2020). We have added the accepted manuscript as supplementary material for your review. Considering our system setup and calibrations carried out for the SP2, we believe the rBC concentrations found in this work are solid results, as described with more details along the text.*

*We made several improvements to the manuscript based on the reviewer's suggestions. First, we opted to remove the 2008 trace element records from the dating section. Although the 2015 and 2008 cores presented an overlap of 7 years (2002-2008), there is a 20% distortion between them in order to match both. This raised an issue about using a core 850 m away, with topographical differences, to date the rBC core. We decided this brings more uncertainty to our work than using the 2015 trace element record to constrain the dating down to 2002, and then base the rest of the dating on the relations of rBC, Na, Sr and S observed for the 2015-2002 period.*

*We also removed the sodium record from the spectral analysis and comparison of rBC and Na transport, as the 2015-2002 Na record is too short to show cycles in the spectrum.*

*On the other hand, we added atmospheric transport simulations using the HYSPLIT model to identify BC source areas. Results corroborated our initial conclusions of Australia and New Zealand as the most probable sources of BC to the TT07 drilling site, and indicated limited influence of South American air masses. More information is presented at the end of this document, after responses to reviewer's suggestions.*

*We also added a comparison of the Antarctic rBC records with snow accumulation, elevation and distance from open sea. In East Antarctica, rBC concentrations have a negative correlation with snow accumulation and positive correlation with elevation and distance to the sea, whereas in West Antarctica rBC concentrations present a positive*

*correlation with snow accumulation and a negative correlation with elevation and distance to the sea.*

*These opposite trends may indicate differences in rBC transport to East and West Antarctica. While for East Antarctica upper tropospheric transport and dry deposition may be the main controllers of rBC concentrations (Bisiaux et al., 2012b), for West Antarctica rBC concentrations may be modulated by intrusion of air masses from the marine boundary layer, contrary to what was previously suggested (Bisiaux et al., 2012a). Low elevations in West Antarctica facilitates the intrusion of moisture-rich cyclones and the transport of aerosols inland (Neff and Bertler, 2015; Nicolas and Bromwich, 2011), while the positive relationship between West Antarctica rBC concentrations and snow accumulation may indicate rBC to be primarily deposited through wet deposition, being scavenged along the coastal regions were snow accumulation is higher.*

*The rBC record we present in this work is from an unique area – the Pine Island/Institute Glacier Divide, where air masses from the Weddel and Bellingshausen Seas converge (Parish and Bromwich, 2007), and is the highest altitude rBC core collected in West Antarctica. Considering all these improvements and new findings, we believe our manuscript is suitable for The Cryosphere.*

*Please find our responses in italic, while we kept your original comments in normal text.*

Original referee comment:

The paper provides a 47-year ice core record of refractory black carbon (rBC) from West Antarctica, specifically Pine Island Glacier. rBC was analyzed by a Single Particle Soot Photometer. The core was dated to 1968, primarily using seasonality of rBC. BC impacts on snow albedo were modeled using the Snow, Ice, Aerosol, Radiation (SNICAR) model. BC emissions were explored with fire spot inventories and spectral analysis was conducted by the REDFIT method.
With respect to the TC guidelines: 1. The paper provides additional field observations of rBC in snow and ice in a data-sparse region of the cryosphere. Making field observations like this available to the community is important for refining our understanding of impurities in the cryosphere and their impact on surface albedo of the Antarctic ice sheet. 2. While the paper provides an additional valuable dataset, it is unclear to me whether the record interpretation is particularly novel. 3. The main finding appears to be that BC transport to the site is not related to marine air masses, which has previously been shown in other ice core records in Antarctica (i.e. Bisiaux et al., 2012). 4. The analytical details appear to be outline in Marquetto et al., 2019, however, I am having trouble locating the manuscript. 5. Thus, I have some remaining analytical questions outlined below. 6. Having access to Marquetto et al., 2019 would assist with reproducibility. 7. The authors provide credit to related work, but I think they should further identify/emphasize the novelty of their contribution. 8. The title clearly reflects the content of

the paper. 9. The abstract provides a concise and complete summary of the existing manuscript. 10. The paper could have benefited from more thorough proofreading before submission; there are some typos. 11. Please refer to 10. 12. Black carbon and refractory black carbon are abbreviated at times and then spelled out at others (i.e. Lines 36, 67, 338). 13. Current figures and tables seem to appropriately support the text. 14. The number of references seems appropriate. 15. The Marquetto et al., 2019 paper would have been useful supplementary information.

*Responses*
*4: Marquetto et al. (2019) was accepted for publication in October 2019 and will be available online in April 2020 in "Advances in Atmospheric Sciences" as Marquetto et al. (2020). The accepted manuscript has been added as supplementary material.*
*5: Along with providing the Marquetto et al. (2020) file, we added additional analytical information in the manuscript based on your suggestions, as is detailed below.*
*6: Marquetto et al. (2020) is available as supplementary information.*
*7: We improved our manuscript, as already mentioned above. Additional information is given at the end of this document, after responses to the reviewer.*
*12: We are now consistent with using abbreviated versions.*
*15: Marquetto et al. (2020) is available as supplementary information.*

**Specific Suggestions:**

Line 29: Typo: 'while there they change'

*Changed the sentence to: "BC particles stay in the atmosphere for just one week to 10 days (Bond et al., 2013; Ni et al., 2014), but  during that time they change the direct radiative forcing..."*

Line 45 – 48: Sentence could be restructured for clarity.

*Changed the sentence to: "More ice core records are needed to understand the spatial variability of BC transport and deposition to Antarctica, as well as to improve general circulation models (Bisiaux et al., 2012b). In this work we add another high-temporal-resolution rBC record from a West Antarctic snow and firn core to the existing literature, in order to contribute to the understanding of BC temporal and spatial variability in Antarctica.*

Line 58 – 61: This section could be expanded, including more specific references.

*Added more information in this section, see below in red color:*

*The core (TT07) was drilled in the 2014-2015 austral summer on the Pine Island Glacier (West Antarctica) at 79°55'34.6"S, 94°21'13.3"W (elevation 2122 m above sea level – a.s.l.), near the Mount Johns Nunatak (located 70 km NE of the drilling site) (Fig. 1) and close to the Institute/Pine Island ice divide. The drilling site was chosen due to its relatively high accumulation rate, that ensures a well preserved seasonal stratigraphic record (Schwanck et al., 2016; Thoen et al., 2018), as well as the region being influenced by air masses from the Weddel, Amundsen and Belingshausen seas (Parish and Bromwich, 2007; Thoen et al., 2018).*

*The West Antarctic Ice Sheet (WAIS) presents lower elevation and lower coastal slopes than the East Antarctic Ice Sheet (EAIS), which facilitates the intrusion of moisture-rich cyclones to the interior of the continent and the transport of aerosols inland (Neff and Bertler, 2015; Nicolas and Bromwich, 2011). Katabatic winds at the drill site are not as strong as they are in most of West Antarctica, due to the higher site elevation compared to the surrounding region (Parish and Bromwich, 2007). Seasonal differences in atmospheric transport have been reported for the TT07 drilling site, with particle trajectories during the austral summer being slow moving and more locally influenced, while during the winter, air trajectories are influenced by oceanic air masses due to strong westerlies. The majority of air masses arrive from the Amundsen Sea and, secondarily, from across the Antarctic Peninsula and Weddell Sea (Schwanck et al., 2017). These are also the preferred pathway for dust particles (Neff and Bertler, 2015).*

Line 77: I cannot find Marquetto et al., 2019 online. Thus, a lot of important analytical details seem to be missing from this manuscript. For example, how long before analysis were the samples melted?

*Marquetto et al. (2020) has been added as a supplement. We added melt time information in the first paragraph of section 3.3: "Samples were melted at room temperature or in a tepid bath not exceeding 25°C, sonicated for 15 min, and then analyzed with one hour of melting."*

Lines 83 and 90: Duplicate sub-section titles.

*Corrected. Section 3.3 should be entitled: "Laboratory and vial cleaning".*

Line 87: Why were polypropylene vials used instead of glass vials? Was particle loss explored with leaching on the vials? Additionally, how long did the samples sit in the vials before analysis?

*Polypropylene vials are widely used for rBC analysis in the SP2. Previous work has tested polypropylene vs glass, and found that polypropylene vials are as suitable to glass*

*unless the sample is left in the liquid state for an extended period of time* (Wendl et al., 2014)*, which was not the case in this study. Samples were melted shortly after analysis.*

Line 146: I don't think Sr is mentioned in Legand and Mayewski, 1997.

*Corrected and added bibliography for Sr, as Legrand and Mayewski (1997) cite only Na. The full sentence is now: "Na and Sr also peak in the dry season (during winter) due to intense atmospheric circulation and transport (*Legrand and Mayewski, 1997; Schwanck et al., 2017*). Increased marine biogenic activity reflects an increase in S in late austral summer (*Schwanck et al., 2017; Sigl et al., 2016*) .*

Line 197: Suggest 'fit' as opposed to 'fitted'.

*Suggestion accepted, thank you.*

Line 199: Suggest using the same past tense, 'chose' as opposed to 'choose'.

*Suggestion accepted, thank you.*

Section 4.1 Dating: Given that the main findings of the paper rely on dating based on seasonality of the rBC record, I think this section could be expanded. For example, the authors could add more discussion as to why the authors think the addition of the rBC record to the layer counting would lead to a dating difference of one year or more, with respect to the core collected nearby that was analyzed for trace elements.

*We opted to use only the 2015 trace element record for dating and constrain the rest of the dating based on the rBC well defined seasonality for West Antarctica (Arienzo et al. 2017; Bisiaux et al., 2012; Winstrup et al., 2017) and for the Pine Island Glacier (Pasteris et al. 2014).*

*To improve our dating for these first meters we reviewed sample resolution for the rBC and trace element cores, and added an additional parameter to dating: the maxima in the non-sea-salt sulfur to sodium (nssS/Na) ratio, a robust seasonal indicator that peaks around the new year (Arienzo et al. 2017). This parameter helps in the identification of the annual layers more than the Na and S records alone. Non-sea-salt sulfur was calculated using Eq. 3 to 6 from Schwanck et al. (2017) and references therein.*

*We consider this dating to have ±2 years uncertainty. The first uncertain year is located at 6.18 m (between 2003 and 2002, figure 2a), where S and nssS/Na peak but no full cycle is observed in the rBC record. We did not consider this to be a year, as rBC does not present a full cycle. The second uncertain year is located at 18.14 m (year 1973,*

*figure 2b) where there is no clear rBC peak but snow accumulation would be anomalously high if considered to be only a year instead of two. We consider this to be an annual pick and consequently two years, as there is no evidence of higher-than-normal snow accumulation in the region for this period (Kaspari et al., 2004).*

[Figure]

**Figure 2. (a) Dating of the snow and firn core based on rBC and using S, Sr, Na and nssS/Na records from nearby core (see section 3.6) as support for the first 6.5 meters. Dashed lines indicate estimated New Year and red dotted line indicate uncertainty in dating, explained in the text. (b) Dating for the full core (y axis logarithmic). Red dotted line indicates uncertainty in dating, as explaned in the text.**

Line 338: The starting date here (1969 – 2015) does not match the abstract or Table 4 (1968 – 2015).

*Corrected the date to 1968-2015.*

Figure 4 Legend: Suggest (bottom) instead of (base).

*Suggestion accepted, thank you.*

Additional information regarding HYSPLIT model:

We added atmospheric transport simulations from using the HYSPLIT model to identify BC source areas. We added two additional sections, one in methodology and another in results. They are as follows:

3. Methodology

[revised manuscript text omitted]